# In Vitro Model to Investigate Communication between Dorsal Root Ganglion and Spinal Cord Glia

**DOI:** 10.3390/ijms22189725

**Published:** 2021-09-08

**Authors:** Junxuan Ma, Vaibhav Patil, Abhay Pandit, Leo R. Quinlan, David P. Finn, Sibylle Grad, Mauro Alini, Marianna Peroglio

**Affiliations:** 1AO Research Institute Davos, 7270 Davos, Switzerland; junxuan.Ma@aofoundation.org (J.M.); sibylle.grad@aofoundation.org (S.G.); mauro.alini@aofoundation.org (M.A.); 2CÚRAM SFI Research Centre for Medical Devices, National University of Ireland Galway, H91 W2TY Galway, Ireland; V.PATIL1@nuigalway.ie (V.P.); abhay.pandit@nuigalway.ie (A.P.); leo.quinlan@nuigalway.ie (L.R.Q.); david.finn@nuigalway.ie (D.P.F.)

**Keywords:** inflammatory cytokine, low back pain, in vitro model, dorsal root ganglion, spinal cord, mixed glial culture, 3R principles

## Abstract

Chronic discogenic back pain is associated with increased inflammatory cytokine levels that can influence the proximal peripheral nervous system, namely the dorsal root ganglion (DRG). However, transition to chronic pain is widely thought to involve glial activation in the spinal cord. In this study, an in vitro model was used to evaluate the communication between DRG and spinal cord glia. Primary neonatal rat DRG cells were treated with/without inflammatory cytokines (TNF-α, IL-1β, and IL-6). The conditioned media were collected at two time points (12 and 24 h) and applied to spinal cord mixed glial culture (MGC) for 24 h. Adult bovine DRG and spinal cord cell cultures were also tested, as an alternative large animal model, and results were compared with the neonatal rat findings. Compared with untreated DRG-conditioned medium, the second cytokine-treated DRG-conditioned medium (following medium change, thus containing solely DRG-derived molecules) elevated CD11b expression and calcium signal in neonatal rat microglia and enhanced Iba1 expression in adult bovine microglia. Cytokine treatment induced a DRG-mediated microgliosis. The described in vitro model allows the use of cells from large species and may represent an alternative to animal pain models (3R principles).

## 1. Introduction

Low back pain (LBP) is often associated with intervertebral disc (IVD) degeneration [1]. Degenerative IVD is characterized by increased levels of pro-inflammatory cytokines [2], such as tumor necrosis factor (TNF)-α, interleukin (IL)-1β, and IL-6, which are observed in disc tissue from discogenic pain patients [2,3,4]. These pro-inflammatory cytokines in degenerative IVD are widely acknowledged as key pain mediators [5]. For instance, serum IL-6 levels are found to be raised in patients with LBP [6]. It has also been shown in a rat model of IVD degeneration that TNF-α injection in IVD accelerated pain-related behavior [7], while an early anti-TNF-α treatment reduced pain-related behavior to levels comparable with those in sham controls [8].

The dorsal root ganglion (DRG) contains the cell bodies of primary sensory neurons [9], which transduce nociceptive stimuli into electrical action potentials (AP), which can be transmitted from the periphery (DRG) to the central nervous system [10] (spinal cord and brain). This process can be affected by pro-inflammatory cytokines and develop a maladaptive response for producing nociceptive or pain signals [11]. For instance, TNF-α sensitizes cultured DRG neurons by dose-dependently enhancing voltage gated sodium currents [12]. The application of IL-1β on dissociated rat DRG neurons decreased the AP voltage threshold and increased AP firing frequency [13]. IL-6 was shown to potentiate spontaneous discharge and response to capsaicin, bradykinin, prostaglandin E2 (PGE2), heat, and electrical field stimulation in primary adult mouse DRG neurons [14].

Apart from neurons, glia in the spinal cord are increasingly recognized as dynamic modulators in pain chronification [15,16]. Gliosis, which is characterized by morphological, molecular, and functional changes, represents an active state of glia [15]. In the spinal cord, microglia and astrocytes are the two glia most documented for their roles in pain [15]. Microgliosis is characterized by a morphological change from ramified to amoeboid shape and an upregulation of microglial markers, such as cluster of differentiation molecule (CD11)-b [15]. Astrogliosis is associated with hypertrophy and upregulation of glial fibrillary acidic protein (GFAP) [17]. Functionally activated microglia and astrocytes display an increased intracellular calcium signal responsible for the calcium-dependent release of neuronal activating factors [18,19]. Gliosis has been observed in multiple animal models of chronic pain [20,21,22], and it has been characterized in post-mortem spinal cord tissue from patients suffering from chronic pain [23]. In animal back pain models [24,25,26] and neuropathic pain model associated with IVD degeneration [27], a higher inflammatory cytokine expression in IVD is usually accompanied by a gliosis in spinal cord. Interestingly, inhibitors of both microglia and astrocytes attenuated pathological pain-related behavior in animal pain models, whereas the normal nociceptive response was not influenced [28,29], indicating the glial cells as a potential target for treating pathological pain without affecting normal pain perceptions.

Since gliosis is important in mediating chronic and pathological pain, the creation of a relevant model is important for understanding the molecular mechanism and discovery of novel drugs. It was found that the electrical stimulation of intact adult rat afferent fibers is enough to activate microglia in the spinal cord [30]. However, the specificity of this in vivo DRG–glia communication can be influenced by other systems (e.g., immune or circulatory system). Additionally, most in vivo models are based on rodents, which in many aspects different from humans [31].

To address these questions, we designed an in vitro experimental set up in which the communication between DRG and spinal cord glia can be specifically and easily accessed. In the in vitro model, spinal cord glial culture was stimulated by conditioned medium of DRG cells. Activation of the microglia and astrocytes was evaluated. The hypothesis of this study is that under the influence of pro-inflammatory cytokines, DRG can release factors that lead to spinal cord glia activation. For the experimental convenience, neonatal rat cells were firstly investigated in the model as a pilot study. Additionally, the model was tested using adult large animal cells (adult bovine DRG and spinal cord).

## 2. Results

### 2.1. Inflammatory Cytokines Enhanced Spontaneous Calcium Response in DRG Neurons

The inflammatory cytokine cocktail (TNF-α, IL-1β, and IL-6, each 10 ng/mL) enhanced spontaneous calcium response (Figure 1D,E) in the DRG neurons. Viable neurons were distinguished from other types of cells by a depolarization-triggered acute and significant rise in intracellular calcium (depolarization chemically activated by increasing external K^+^) (Figure 1A,B).

Quantification of the calcium transients showed that for cytokine cocktail-treated DRG neurons, the proportion of viable neurons displaying spontaneous events was increased by 38.5% (neurons with calcium events larger than the threshold, median level from 72% to 99.7%), and maximum fluorescent elevation speed (MFES) was increased by 77.6% (median level from 0.29 to 0.52) (Figure 1G). Although peak frequency over 300 s was decreased (median level from 5 to 2) (Figure 1H), peak height was increased two-fold (median level from 0.31 to 0.61) (Figure 1I). When the data of different experiments using different donors were separately investigated, proportions of neurons with spontaneous responses and peak height were increased in inflammatory cytokine-treated DRGs compared with the untreated group in all donors (Figure 1J,L). For MFES, the same trend was observed in two out of the three experiments using different donors (Figure 1K).

### 2.2. Inflammatory Cytokines Did Not Influence CGRP Expression in Neonatal Rat DRG Cells

The neurotransmitter calcitonin gene-related peptide (CGRP) is known to play a major role in the pathogenesis of pain syndromes [32]. Thus, the expression of CGRP in neonatal rat DRG was evaluated using an immunofluorescent method (Figure 2A,B). Inflammatory cytokine treatment for two days decreased the proportion of neurons positive to CGRP (Figure 2C). The results from three independent experiments using different donors showed the same trend (Figure 2D). No significant difference in CGRP staining intensity relative to tubulin β-III (TUBB3) was detected (Figure 2E,F).

### 2.3. Inflammatory Cytokines Induced a DRG-Mediated Microgliosis in Mixed Glial Cuture of Neonatal Rat Spinal Cord

Gliosis in the spinal cord is increasingly recognized as a key mechanism of pain chronification, and CD11b and GFAP-expressing levels are generally used as biomarkers for microgliosis and astrogliosis, respectively [15]. Neonatal rat DRG and spinal cord glia communication was investigated in vitro using DRG-conditioned medium to stimulate MGC from the spinal cord. The gliosis was evaluated via immunostaining of GFAP and CD11b. An upregulation of CD11b in microglia but not GFAP in astrocytes was observed in MGC stimulated by conditioned medium from cytokine-treated DRG compared with conditioned medium from non-treated DRG.

Overall, the first conditioned medium from cytokine-treated DRG showed the strongest effect on the elevation of microglial CD11b expression compared with the first conditioned medium from non-treated DRG (Figure 3C,D). Indeed, the immunostaining intensity of CD11b in MGC treated with CytCM1 (the first conditioned medium collected from cytokine-treated DRG) was more than two-fold higher than MGC treated with ConCM1 (the first conditioned medium collected from DRG not treated with cytokines) (Figure 3G). This strong effect was observed in all the three experiments using different donors (Figure 3I).

The second conditioned medium from cytokine-treated DRG (CytCM2) upregulated microglial CD11b expression by 36.1% compared with the second conditioned medium from non-treated DRG (Figure 3G). All the donors showed the same trend (Figure 3I). Even after removal of the inflammatory cytokines by medium change, the cytokine-treated DRG cells continued to induce a significantly higher CD11b expression in MGC compared with the non-treated DRG group, thus pointing to the sole effect of DRG–glia communication on microgliosis.

The direct application of cytokines to MGC induced only minimal changes in CD11b staining intensity (+10.7% compared with control without statistical significance) (Figure 3G,I). This suggests that the effect of DRG-conditioned medium is not due to the added cytokines, which were potentially carried over in minimal amounts in the DRG-conditioned medium, but rather due to DRG-derived molecules. Cytokines alternatively increased the CD11b stained area by 32%, which represents a higher cell confluency (Figure 3A,B,H,J).

Unlike the evaluation of microgliosis, the expression of GFAP as a biomarker of astrogliosis was not found to be significantly different among groups (Figure 3K–N).

### 2.4. Inflammatory Cytokines Induced a DRG-Mediated Microglial Shape Change in Mixed Glial Cuture of Neonatal Rat Spinal Cord

Microgliosis in vivo has been characterized as a change from a ramified to an amoeboid shape [15]. In our study, the strongest microglial shape change was observed following exposure to the first conditioned medium from cytokine-treated DRG (CytCM1) compared with the first conditioned medium from untreated DRG (ConCM1) (Figure 4A,B).

Since the first conditioned medium from cytokine-treated DRG still contains the cytokines added to stimulate the DRG, it represents a combined effect of cytokines and DRG-derived molecules on the microglial shape change. The first conditioned medium from cytokine-treated DRG (CytCM1) significantly increased cell solidity (by 25.5%) and reduced microglial cell area (by 51.6%) compared with conditioned medium from non-treated DRG (ConCM1) (Figure 4C,D), and the same trend was observed among different donors (Figure 4G,H). ‘Elliptical form factor’ and ‘Transformation index’ were not significantly different among groups (Figure 4E,F).

The second conditioned medium from cytokine-treated DRG (CytCM2) did not significantly change microglia shape compared with the second conditioned medium from non-treated DRG (ConCM2) (Figure 4C–G). Likewise, direct MGC cytokine treatment (Cyt) did not significantly influence microglial shape compared with basal medium treatment control (Con) (Figure 4C–G). Neither DRG-produced molecules alone (CytCM2 versus ConCM2) nor cytokine alone (Cyt versus Con) led to significant microglial shape change.

### 2.5. Inflammatory Cytokines Induced a DRG-Mediated Astrocyte Arbor Branching Reduction in Mixed Glial Cuture of Neonatal Rat Spinal Cord

Along with microglia shape changes, astrocyte hypertrophy and loss of astrocytic ramification could be observed when stimulated by the first conditioned medium of cytokine-treated DRG (CytCM1) compared with the first conditioned medium of non-treated DRG (ConCM1) (Figure 5A,B). Since astrocytes generally displayed a higher ramified structure than microglia, ‘Sholl’ analysis was used to characterize the pattern of astrocyte processes (Figure 5G).

The first conditioned medium from cytokine-treated DRG (CytCM1) reduced intersection numbers (representing branch frequencies) over the distance from 0 to 40 μm compared with the conditioned medium from non-treated DRG (ConCM1) (Figure 5H). Comparing CytCM1 with ConCM1, the median ‘Maximum intersection number’ was decreased from 8 to 6 (Figure 5I), and the median ‘Sum of intersections’ was decreased from 144 to 134 (Figure 5J). Two out of the three donors showed the same trend regarding the ‘Maximum intersection number’ (Figure 5K), while all the three independent experiments showed the same trend regarding the ‘Sum of intersections’ (Figure 5L).

No significant difference of astrocytic ramification was detected between CytCM2 and ConCM2 (representing the sole effect of DRG-produced molecules) and between Cyt and Con (representing direct effect of cytokines) (Figure 5I–L).

Astrocyte cell solidity, area, ‘Elliptical form factor’, and ‘Transformation index’ were not found to be significantly different among groups (Figure 5C–F).

### 2.6. Inflammatory Cytokines Induced a DRG-Mediated Increase in Spontaneous and ATP-Stimulated Calcium Response in Neonatal Rat Microglia

The regulation of glial calcium signal plays an essential role in the regulation of pain [33]. The spontaneous calcium oscillation and the calcium response to the neurotransmitter adenosine triphosphate (ATP) were found to be enhanced by the conditioned medium from cytokine-treated DRG compared with non-treated DRG in the calcium imaging of MGC. The region of interest (ROI) representing microglia was recognized by performing immunofluorescence of CD11b (biomarker of microglia) following calcium imaging in the mixed culture (Figure 6O,R).

For the spontaneous calcium response in microglia, peak heights were not found to be significantly different following exposure to conditioned medium collected from untreated DRGs or DRGs treated with cytokines (CytCM1 versus ConCM1 and CytCM2 versus ConCM2) (Figure 6A,C). Only the direct application of cytokines on MGC elevated spontaneous calcium peak height in microglia compared with basal medium treatment control (Cyt versus Con) (Figure 6A,C).

The frequency of spontaneous calcium peaks was significantly increased when the MGC was stimulated with the first conditioned medium from cytokine-treated DRG cells compared with the first conditioned medium from non-treated DRG cells (ConCM1 versus CytCM1, median peak frequency 3 to 8) (Figure 6B), but this trend was not consistently observed among all donors (Figure 6D). The second conditioned medium showed the same trend on frequency of calcium peaks (ConCM2 versus CytCM2 median peak frequency also from 3 to 8) (Figure 6B,E,F), and two out of the three donors showed the same trend (Figure 6D). The direct application of cytokines did not influence spontaneous peak frequency (Figure 6B).

For the ATP-induced calcium response of microglia, calcium peak height was increased by 57.5% when comparing the first conditioned medium from cytokine-treated DRG with the first conditioned medium from non-treated DRG (CytCM1 versus ConCM1, Figure 6G), but this trend was not consistently observed among all donors (Figure 6I). The second conditioned medium from cytokine-treated DRG showed a much stronger effect on the calcium peak height, which was 142.2% higher compared with the second conditioned medium from non-treated DRG (CytCM2 versus ConCM2, Figure 6G,K–N,P,Q), with two out of the three donors showing the same trend (Figure 6I). The ATP-induced calcium peak height was not significantly different when comparing direct cytokine treatment with basal medium treatment (Cyt versus Con, Figure 6G).

The ATP-induced calcium flux duration was not significantly different between CytCM1 and ConCM1 (Figure 6H), but it was significantly increased in CytCM2 compared with ConCM2 (median level from 2 to 3 s Figure 6H) showing the same trend among the three donors (Figure 6J); direct cytokine treatment significantly extended the calcium flux duration from 1 to 3 s (Cyt versus Con, Figure 6H), and two out of the three donors showed the same trend (Figure 6J). The enhanced ATP-induced calcium peak height and calcium flux duration by CytCM2 compared with ConCM2 suggest that the microglial calcium response to ATP can be promoted by the DRG–spinal cord glia communication.

### 2.7. Inflammatory Cytokines Induced a DRG-Mediated Increase in ATP-Stimulated Calcium Response in Neonatal Rat Astrocytes

The spontaneous calcium signal in astrocytes was not significantly enhanced by the conditioned medium of cytokine-treated DRG compared with non-treated DRG (CytCM1 versus ConCM1, CytCM2 versus ConCM2), but ATP-stimulated calcium response was elevated by the second conditioned medium of cytokine-treated DRG compared with the second conditioned medium from non-treated DRG (CytCM2 versus ConCM2).

For the spontaneous calcium oscillation in astrocytes, the CytCM1 induced higher calcium peak frequency but lower peak height compared with the first conditioned medium of non-treated DRG (CytCM1 versus ConCM1 median peak frequency from 4 to 11, median peak height decreased by 48.7%) (Figure 7A,B). No significant difference was detected in spontaneous calcium peak height and peak frequency between the second conditioned medium of cytokine-treated DRG and the second conditioned medium of non-treated DRG (CytCM2 versus ConCM2) (Figure 7A,B,E,F). Applying cytokines to the MGC increased the spontaneous peak height (Cyt versus Con, Figure 7A) with all donors showing the same trend, whereas spontaneous peak frequency was not found to be significantly different between Cyt and Con.

For the calcium response to ATP in astrocytes, no significant difference was found between the CytCM1 and ConCM1 groups for both calcium peak height and calcium flux duration. However, peak height and calcium flux duration were all significantly increased (by 218.7% and 50%, respectively) by the treatment of CytCM2 compared with ConCM2 (Figure 7G,H,K–N,P,Q), with all donors showing the same trend for ATP-induced peak height (Figure 7I), and two out of three donors showing the same trend for the ATP-induced calcium flux duration (Figure 7J). Likewise, applying cytokines directly to the MGC increased both calcium peak height and calcium flux duration (Cyt versus Con by 282.8% and 100%, respectively, Figure 7G–J) with all donors showing the same trend in ATP-induced peak height (Figure 7I) and two out of three donors showing the same trend for the ATP-induced calcium flux duration (Figure 7J). The higher ATP-induced calcium peak height and flux duration by CytCM2 compared with ConCM2 suggest the role of DRG in mediating enhanced astrocytic calcium response to ATP.

### 2.8. Inflammatory Cytokines Had Limited Effect on Calcium Signal of Bovine DRG Neurons

Adult bovine DRG cell culture was tested in the model. Immunofluorescence showed that the bovine DRG cell culture contains neurons (TUBB3-stained cells in Figure 8A,B), satellite cells (GFAP-stained cells in Figure 8A,B), and macrophages (Iba1 (ionized calcium binding adaptor molecule)-stained cells in Figure 8B). Schwann cell biomarker SOX10 [34] was not detected, indicating that the mixed culture may not contain Schwann cells (Figure 8A). In the DRG culture, most smaller cells were identified as satellite cells. Larger cells were recognized as neurons with soma diameter 30–100 μm (Figure 8A), and most of the culture surface was covered by the neurite outgrowth (Figure 8B).

Different from neonatal rat, the calcium signal of bovine DRG neurons was only slightly influenced by the treatment of cytokines (IL-1β, IL-6, and TNF-α all at 10 ng/mL) (Figure 8C,D). MFES and peak height were increased by only around 5% and 4%, respectively comparing the cytokine-treated group with the non-treated control (Figure 8F,H). No statistical significance was detected for this small difference. The proportion of cells with spontaneous calcium response and calcium peak height were not found to be different comparing the cytokine-treated group with the non-treated control (Figure 8E,G).

### 2.9. Iba1 Expression of Bovine Microglia Was Upregulated by Cytokine-Treated Bovine DRG Cells

Bovine cervical spinal cord was dissected for glial culture (Figure 9A,B). Microglia was labeled by anti-Iba1 primary antibody (Figure 9C,D). Although cytokine treatment only marginally enhanced calcium signals within DRG neurons, it was enough for the second DRG-conditioned medium to trigger Iba1 upregulation in spinal cord microglia comparing with the non-treated control (CytCM2 versus ConCM2). On average, the Iba1-stained area was elevated by 49.4%, and Iba1 staining intensity was increased by 39.6% comparing conditioned medium from cytokine-treated DRG cells with non-treated DRG cells (Figure 9C–F).

Calcium signal in bovine microglia was not significantly influenced by the secondly collected DRG-conditioned medium. Neither spontaneous nor ATP triggered calcium response showed significant difference comparing cytokine-treated DRG cells with the non-treated control (CytCM2 versus ConCM2) (Figure 9G–L).

## 3. Discussion

It is known that inflammatory cytokines are involved in IVD matrix degradation [35], DRG nerve ingrowth [36], and sensitization [14,37], which are all associated with IVD degeneration and discogenic low back pain. What is incompletely understood is how the pain becomes chronic. It has been shown that glial cells in the spinal cord play important roles in the maladaptive regulation of the central nervous system (central sensitization) and contribute to pain chronification [15]. The communication between peripheral DRG and the central spinal cord glia may explain why DRG, which are exposed to cytokines released from degenerative, inflamed IVDs, can lead to central glial activation in the spinal cord [38].

In this study, we characterized the DRG–glia communication in an in vitro model and found that neonatal rat DRG cells, when exposed to the inflammatory cytokine cocktail (TNF-α, IL-1β and IL-6), communicate with glial cells from the spinal cord and contribute to glial activation. Specifically, we found that spontaneous calcium oscillation in neonatal DRG neurons was enhanced by the cytokine treatment, which may indicate a higher ectopic firing and sensitization of these neurons. Although calcium imaging is an indirect approach to evaluate neuronal discharge, a correlation between calcium transients and electrical action potential has been evidenced in former studies comparing calcium imaging to electrophysiological methods such as patch clamp [39]. Furthermore, the calcium signal within DRG neurons is essential for neurotransmitter release and pain/nociceptive signal transmission in the dorsal horn of the spinal cord [40]. These calcium imaging data of DRG cells are in agreement with former electrophysiological studies [12,13]. The finding that the nociceptive neurotransmitter CGRP was not upregulated in the DRG cells following cytokine treatment could be explained by the fact that 12 h of cytokine treatment was not long enough to induce CGRP upregulation. Indeed, in a former study using an inflammatory rat pain model, immunoreactive CGRP level was first decreased within two days, but then, it increased compared with control at eight days [41].

Under the stimulus of cytokine-treated DRG conditioned medium, both microglia and astrocytes dissociated from neonatal rat spinal cord showed enhanced calcium oscillations observed by calcium imaging. Particularly, the ATP-stimulated calcium response within both microglia and astrocytes was elevated by cytokine-treated DRG cells compared with non-treated DRG cells. When evaluating biomarkers of gliosis, CD11b upregulation (relating to microgliosis) was observed under the influence of DRG cells treated with the cytokines, but no upregulation of astrocytosis biomarker GFAP was detected. This can be explained by a former in vivo study where the upregulation of microgliosis biomarker could be readily observed from day one to seven, whereas astrocytosis biomarker upregulation only initiated from day four to seven [42]. Astrocytosis biomarker GFAP may only show an upregulation in a later phase (more than three days), while in our in vitro model, the MGC was treated with DRG-conditioned medium for only 24 h.

Although GFAP expression remained unchanged, astrocytes already displayed an activated state as attested by the enhanced calcium response to added ATP using calcium imaging analysis. Calcium imaging seems to be a more sensitive method in the early detection of astrocytic activation than GFAP expression. Calcium signaling within glia is of important relevance for its involvement in the regulation of pain pathway. Indeed, the glial release of many pain mediators and neurotransmitters is regulated by the oscillation of intracellular calcium level [33,43]. The mechanism of the increased calcium response to ATP within astrocytes may involve an upregulation of purinergic receptors [44] and needs further investigation.

Only the first conditioned medium of cytokine-treated-DRG significantly increased the solidity of neonatal rat microglia and decreased the branching complexity of astrocytes. This conditioned medium contained both the DRG-derived molecules and the inflammatory cytokines, so it seems that the structural plasticity of glia only occurs when the direct and indirect (via DRG) effects of inflammatory cytokines are combined. However, one limitation of this study on cell shape is that primary cell culture in a culture dish is different from a three-dimensional matrix in vivo, and more advanced organotypic cultures should be considered in future studies.

Following the use of neonatal rat cells in the model, cells from large species (adult cattle) were tested for three reasons: (1) the nervous system is still developing in neonatal rats, and the nociceptive system is different compared with adults [45]; (2) the culture of neonatal DRG cells requires supplementary nerve growth factor (NGF), which could have a priming effect on nociceptors; (3) DRG cells from rodents were found to be different compared with human DRG tissue in terms of size and gene expression profile [46].

Due to limited access to human DRG tissue, DRG from large animals is a promising alternative. For example, DRG neuronal subtypes were classified in non-human primate based on the transcriptome [47]. Unlike macaque, bovine tissue can be easily accessed from abattoir without additional animal euthanasia. The culture of bovine DRG cells has been reported [48] but has not been widely used in the study of pain-associated biology. The gene expression profile of bovine DRG needs to be compared with human in the future to support the use of bovine DRG culture.

In our primary bovine DRG cell culture, the diameter of neuronal soma ranged between 30 and 100 μm, which is close to the reported human DRG cell culture [49]. Similar to neonatal rats, after being stimulated by inflammatory cytokines, bovine DRG-conditioned medium also activated microglia isolated from spinal cord, as shown by the upregulation of microglial Iba1. Thus, the communication between DRG and spinal cord glia observed in neonatal rat cells was also identified in adult bovine cells. On the other hand, differences were observed between neonatal rat and adult cattle. Spontaneous calcium signaling of bovine DRG neurons was less influenced by the cytokines compared with neonatal rat DRG neurons. While calcium response to ATP in neonatal rat microglia was enhanced by cytokine-treated DRG-conditioned medium, this response of bovine microglia was only marginally elevated. Overall, studies performed in neonatal rats did show approximation to large animals, but there were also differences. The differences can be due to (1) a changing nociceptive system during development (neonatal versus adult) [45] and (2) a difference between species (rat versus calves) [48].

Three facts suggest the implication of DRG-derived molecules in mediating glial activation: (1) The medium change after collecting the first DRG-conditioned medium largely removed the manually added cytokines. Nonetheless, the second DRG-conditioned medium still significantly upregulated CD11b expression in microglia; (2) As a control, applying cytokines directly to MGC did not significantly upregulate CD11b expression in microglia and did not affect shape in both microglia and astrocytes. This indicates that glial activation cannot be solely attributed to cytokines carry-over in the DRG-conditioned medium; (3) The second DRG-conditioned medium showed a stronger effect on increasing ATP-induced calcium signal in both microglia and astrocytes compared with the first DRG-conditioned medium. Based on these facts, the interference from the carry-over of minimal amounts of cytokines from the first to the second conditioned medium can be excluded.

Another question is whether the in vitro set up (conditioned medium stimulation) is relevant to in vivo conditions. Recently, a direct cerebrospinal fluid exchange between DRG and spinal cord has been validated in vivo, proving that the typical synaptic transmission is not the only method of DRG–spinal cord communication [50]. Indeed, intrathecally administrated fluoro-emerald could be detected in the cells of DRG capsule, satellite glial cells, interstitial space, as well as in small and medium-sized neurons, indicating that spinal cord cells are exposed to molecules released from both neurons and glia in DRG [50]. In this regard, our in vitro model can be representative, but has limitations regarding not recapturing the whole in vivo complexity considering the existence of both synaptic transmission and cerebrospinal fluid exchange.

The in vitro model described in this study could be used to investigate molecular mechanisms and thereby contribute to reduce the need for in vivo pain models in line with 3R principles. For instance, in vivo neuropathic pain studies have already suggested that molecules such as matrix metallopeptidase (MMP)-9, MMP-2, C-X3-C motif chemokine ligand 1 (CX3CL1), C-C motif chemokine ligand 2 (CCL)-2, and CCL-21 are involved in the communication of DRG and spinal cord glia [51,52,53,54,55]. The next step is to investigate what exactly the molecules released from the DRG are that activate the spinal cord glia and serve as potentially novel targets of chronic pain treatment. However, we are aware that findings from in vitro cannot represent the perception of pain; validations from animal model or clinical studies are necessary.

## 4. Materials and Methods

### 4.1. Cell Source and Study Design

Primary cultures of both DRG and MGC were obtained from the whole spine of neonatal rats (10–12 days old, *n* = 7–9 for each independent experiment. Sex of the rats was randomly distributed between groups). The animals used in the study were approved by the Animal Care Research Ethics Committee (ACREC) of the National University of Ireland, Galway, and under appropriate individual and project authorizations from Health Products Regulatory Authority (HPRA) of Ireland (License number B342), and the procedures were performed in accordance with the Principles of Laboratory Animal Care.

The study design included 2 parts: (1) The pro-inflammatory cytokines (IL-1β, IL-6, and TNF-α all at 10 ng/mL) were applied to the primary neonatal DRG cells. The calcium imaging was recorded in the DRG neurons, and neurotransmitter calcitonin gene-related peptide (CGRP) expression was analyzed using immunofluorescence; (2) To study DRG–glia communication, the DRG-conditioned media were collected to stimulate spinal cord MGC following pre-treating the DRG using the pro-inflammatory cytokine cocktail. The purpose was to investigate whether glia could be activated under the influence of DRG-produced molecular signal in the conditioned medium. Since the first collected conditioned medium also contained the added cytokine cocktail and thus did not represent the sole effect of DRG–spinal cord glial communication, a second conditioned medium was collected to stimulate MGC after medium change following the first conditioned medium collection where the initially added cytokines were removed.

The activation of the glia in the MGC was assessed using immunofluorescence to evaluate the expression level of CD11b as a microgliosis marker and glial fibrillary acidic protein (GFAP) as an astrogliosis marker. Shape analysis provided another indicator of gliosis. Functional activation of the glia cells was investigated using calcium imaging. Both spontaneous calcium oscillation and calcium response to neurotransmitter ATP were analyzed.

### 4.2. Isolation and Culture of Primary DRG Cells Dissociated from Neonatal Rats

The DRG cell dissociation and culture were modified based on previous reports [56,57]. Briefly, DRG were dissected from whole spines of neonatal rats (Sprague–Dawley rat, 10–12 days old, 8–9 rats for one independent experiment, three independent experiments were performed). The spines were harvested and then split longitudinally into 2 parts in an anterior–posterior direction. All DRG from whole spines were dissected and stored in ice-cold phosphate buffered saline (PBS). Then, the DRG were digested with 4 mL of 0.25% trypsin/EDTA (Gibco, Paisley, UK) at 37 °C for 30 min on an orbital shaker. To stop the enzymatic action of trypsin, 200 μL of fetal calf serum (FCS, Sera Plus, Biotech, Aidenbach, Germany) were added. The mechanical dissociation was performed by trituration of the loosened DRG with a pipette tip (P1000 and P200) until the solution became turbid. The cell suspension was filtered through a 70 μm cell strainer (Falcon, Corning, Tewksbury, MA, USA), the cells were pelleted down by centrifuging at 800 rpm for 10 min and re-suspended in 2 mL of α-MEM (HyClone™, Northumberland, UK) with 10% FCS, 1% penicillin/streptomycin (PS, Gibco, Paisley, UK), and 10 ng/mL nerve growth factor (NGF, R&D, Minneapolis, MN, USA). Cells were plated at a density of 10,000 cells /cm^2^ in 8-well chambers (Ibidi, Gräfelfing, Germany) (in 100 μL medium per well) coated with poly-D-lysine (PDL, Sigma-Aldrich, Buchs, Switzerland); coating was performed by incubating with 100 μg/mL PDL for 1 h at 37 °C and washing with PBS three times. The cells were cultured at 37 °C and 5% CO_2_ for two days before being treated with pro-inflammatory cytokines.

### 4.3. Pro-Inflammatory Cytokine Treatment

According to preliminary tests [58], a concentration of at least 10 ng/mL was required to significantly activate spinal glia in vitro when exposed to either of the cytokines IL-1β, IL-6, or TNF-α, and the combination of the three cytokines at 10 ng/mL each had the strongest effect. Since multiple cytokines concurrently act in vivo, an inflammatory cocktail was created by adding recombinant rat IL-1β, IL-6, and TNF-α (all purchased from R&D, Minneapolis, MN, USA) at a final concentration of 10 ng/mL each into serum-free α-MEM containing 1% PS and 10 ng/mL NGF. For the cytokine-treated group, the media of primary DRG cells were replaced by medium containing inflammatory cocktail two days after cell plating. As control, media of primary DRG cells were changed to the basal media (α-MEM with 1% PS and 10 ng/mL NGF) without inflammatory cytokines. The treatment lasted for 12 h, since a longer time exposure of DRG cells to serum-free medium reduced cell viability.

To study the prolonged neuropathy of DRG neurons, which can last even after the inflammatory cytokines are removed, media of both groups were changed to basal media without cytokines after 12 h of inflammatory cocktail treatment, and both groups were maintained in basal medium for another 12 h before performing calcium imaging analysis.

### 4.4. Calcium Imaging of Primary Neonatal Rat DRG Cells

The calcium imaging method is based on former literature [59,60]. Briefly, DRG cells were washed with Krebs–Ringer solution (NaCl 119 mM, KCl 2.5 mM, NaH_2_PO_4_ 1.0 mM, CaCl_2_ 2.5 mM, MgCl_2_ 1.3 mM, HEPES 20 mM, and D-glucose 11.0 mM dissolved in ion-free water and filtered with 0.22 μm filter) and then incubated with 5 μM Fluo-4-AM (ThermoFisher, Bleiswijk, The Netherlands) dissolved in Krebs–Ringer solution supplemented with 0.3% bovine serum albumin (BSA, Fluka, Rochester, NY, USA) for 40 min at 37 °C. Cells were washed once with Krebs–Ringer solution and further incubated for 10 min at 37 °C to de-esterize Fluo-4-AM. Following this step, cells were washed again and then preserved in 180 μL Krebs–Ringer solution. Time-lapse images were acquired using an Olympus inverted microscope (IX81, Tokyo, Japan) at 10× magnification, 0.75 numerical aperture, and 0.6 mm working distance at an excitation of 472/30 nm and an emission at 520/35 nm. The image acquisition rate was set at one image per second, and the recording time was set to 350 s. The first 300 s were taken as baseline level and were used to study spontaneous calcium response. To identify neurons from other types of cells, at the 300th second, 20 μL of 500 mM potassium chloride (final concentration at 50 mM) was added to the DRG cells, and recording was continued for another 50 s.

The calcium imaging analysis was modified based on previously reported protocols [60,61,62] and included three steps: (1) definition of the region of interest (ROI), (2) normalization of the fluorescent change over time, and (3) analysis of the calcium fluorescent peaks. ROI was established using the circle manual drawing tool and the ‘ROI manager’ within ImageJ Fiji (v1.52p, NIH, Bethesda, MD, USA) in the 8-bit stack image produced by ‘Z project’ function from 1 to 300 s with the projection type set as ‘average intensity’. The normalization and calcium peak analysis was performed using ‘R’ (v3.6.2, Lucent Technologies, Murray Hill, NJ, USA) based on a previously reported methodology [60]. Briefly, to exclude the interference of Fluo-4 loading efficiency, the fluorescent intensity (F) over time was normalized using the baseline fluorescent level averaged from 0 to 300 s (F_0_) by the equation (F − F_0_)/F_0_. The peak analysis was performed using the derivative method. The first derivative (d∆F/F_0_) of the ∆F/F_0_ curve was used to represent the speed of the intracellular calcium concentration change over time. To exclude noise, a spontaneous calcium peak was defined as maximum d∆F/F_0_ higher than 95% of all peaks. The proportion of ROIs responding, maximum fluorescent elevation speed (MFES), peak frequency, and peak height were calculated during 0–300 s to evaluate spontaneous response. Since only neuronal structures immediately depolarize due to extracellular increase in potassium chloride, ROIs obtained for the primary DRG cells were subdivided to those with a potassium-induced calcium peak (DRG neuronal structures) and those without a potassium-induced peak (non-neuronal structures) after adding potassium chloride (potassium chloride-evoked calcium peak height larger than 95% of maximum peak height among all ROIs). One of the cultures was fixed in 4% paraformaldehyde immediately after calcium imaging and was immune-stained with anti-Tubulin β-III (Neuronal bio-marker) antibody (Detailed method in Section 4.6) to validate that the potassium added could efficiently differentiate neurons from other types of cells (Figure 1A–C).

### 4.5. Neonatal Rat DRG Cell-Conditioned Medium Preparation for MGC Stimulation

The methods for the neonatal rat DRG cell dissociation and primary culture were the same as described in Section 4.2. Before collecting DRG, the spinal cord was carefully removed from the spinal canal and preserved in ice-cold PBS for mixed glial culture (MGC) (described in Section 4.7). For each group in each independent experiment (overall three independent experiments), DRG from whole spines of 9–10 neonatal rats were dragged out using fine forceps, combined for the cell dissociation, and seeded in 6–12 wells of a 96-well plate (ibidi, Gräfelfing, Germany) coated with PDL (3–6 technical replicates per group). Two days after cell seeding, the medium was replaced by basal medium supplemented with inflammatory cocktail (IL-1β, IL-6, and TNF-α, all 10 ng/mL) or basal medium again as control. Following 12 h of inflammatory cytokine treatment, the first conditioned media were collected, and the DRG cells in both groups were refreshed with basal medium. After another 12 h, the second conditioned media were collected. The first DRG cell conditioned medium after cytokine treatment still contained the inflammatory cytokine cocktail and represented a combined effect of cytokine and DRG-released molecules, while the second conditioned medium was collected after medium change following cytokine stimulation and thus represented the sole effect of DRG-released molecules (sole effect of DRG-spinal cord glia communication). The conditioned media were collected and stored at −20 °C until used to stimulate MGC.

### 4.6. CGRP Immunofluorescence of the Neonatal Rat DRG Cells

After collecting the second DRG-conditioned medium, DRG cells were fixed in 4% paraformaldehyde (Sigma, Temecula, CA, USA) for 1 h at room temperature and washed with PBS three times. Fixed cells were incubated with blocking solution (PBS with 0.1% Triton-X and 3% BSA) at room temperature for 1 h. The immunostaining was performed using the primary antibodies mouse anti-Tubulin β-III monoclonal antibody (TUBB3, 1:200) (Sigma, Temecula, CA, USA) and rabbit anti-Calcitonin Gene Related Peptide antibody (CGRP, 1: 200) (Sigma, Temecula, CA, USA) diluted in PBS containing 0.1% Triton-X (Sigma, Saint Louis, MO, USA) and 3% BSA. The incubation with primary antibodies was performed at 4 °C overnight, which was followed by an incubation with the secondary antibodies at room temperature for 1 h. The secondary antibodies were donkey anti-mouse AlexaFluor 488 conjugated antibody (1:2000, Thermo Fisher, Eugene, OR, USA) and goat anti-rabbit AlexaFluor 546 conjugated antibody (1:500, Thermo Fisher, Eugene, OR, USA). Nuclei were stained using 0.1 μM Hoechst (Thermo Fisher, Rockford, IL, USA) for 10 min at room temperature followed by two washes with PBS. Omission of the primary antibody served as a negative control. Ten random images per well were acquired using Olympus inverted microscope (IX81, Tokyo, Japan) at 20× magnification, 0.45 numerical aperture and 7.2 mm working distance, at an excitation: 472/30 nm/emission: 520/35 nm for Alex488, excitation: 562/40 nm/emission: 624/40 nm for Alex 546, and excitation: 377/50 nm, emission: 477/60 nm for Hoechst. Three independent experiments were performed using different rat donors, while each independent experiment included 3–6 wells of culture per group. The image analysis was performed using imageJ Fiji. CGRP (the nociceptive neurotransmitter) staining fluorescent intensity was normalized by TUBB3 (general neuronal marker) in each field.

### 4.7. Isolation and Culture of Primary MGC from Neonatal Rat Spinal Cords

The intact spinal cord was obtained using a ‘hydraulic extrusion’ method [63]. The whole spine was dissected out and preserved in ice-cold PBS. The proximal end of a P200 pipette tip was trimmed to firmly fit a 20 mL syringe for proper connection. The tip connected with a 20 mL PBS-filled syringe was inserted into the spinal canal of the distal end of the spine. With the spine straightened, pressure was applied until the spinal cord was extruded into a Petri dish containing sterile PBS. The spinal cord was preserved in ice-cold PBS while dissecting DRG. Under a dissection microscope (Leica, Wetzlar, Germany), the spinal cord meninges tissue was carefully removed. Following mincing with a scalpel, the spinal cord tissue was digested in 0.25% trypsin (1 mL per spinal cord) at 37 °C on an orbital shaker for 15 min. The enzyme was stopped by adding equal amounts of high-glucose Dulbecco’s Modified Eagle Medium (4.5 g/L DMEM, Gibco, Paisley, UK) with 10% FCS and 1% P/S. Mechanical trituration was performed by passing the tissue through 18 G (twice), 21 G (twice), and 25 G (once) needles sequentially. The dissociated tissue was filtered through a 70 μm cell strainer, and cells were pelleted down by centrifugation at 1200 rpm for 10 min. Cells were resuspended in 2 mL DMEM supplemented with 10% FCS and 1% P/S. Cells were seeded at the density of 10,000 cells/cm^2^ in a 96-well-plate (ibidi, Gräfelfing, Germany) coated with PDL. The MGC was maintained at 37 °C and 5% CO_2_; half of the medium was replaced every 2–3 days.

### 4.8. Stimulation of Neonatal Rat MGC by DRG-Conditioned Medium

The primary MGC was stimulated by DRG-conditioned medium at four days after cell seeding for a proper glial cell adhesion and confluency. The culture media of MGC were changed to first and second DRG-conditioned media, both of which were collected from DRG either pre-treated with inflammatory cocktail (‘CytCM1’ and ‘CytCM2’ as first and second collected conditioned media, respectively) or DRG pre-treated with basal media (‘ConCM1’ and ‘ConCM2’ as first and second collected conditioned media, respectively). MGC treated with DRG basal medium (α-MEM with 1% PS and 10 ng/mL NGF) served as negative control (‘Con’ group), and MGC directly treated with inflammatory cocktails (α-MEM with 1% PS and 10 ng/mL NGF supplemented with IL-1β, IL-6, and TNF-α, all 10 ng/mL) was set as positive control (‘Cyt’ group). The treatment lasted for 24 h; then, the MGCs were fixed using 4% paraformaldehyde for 1 h at room temperature and washed with PBS three times for immunofluorescent analysis. In this experiment, each group contained 3–6 wells (technical replicates) in each independent experiment. Three independent experiments were performed using different rat donors (9–10 rats per independent experiment).

### 4.9. Immunofluorescence of Neonatal Rat MGC

The immunofluorescence protocol for the MGC was the same as for the DRG primary culture in the former sections. The primary antibodies were mouse anti-Integrin αM CD11b monoclonal antibody (CD11b, 1:200) (Millipore, Darmstadt, Germany) and rabbit anti-Glial Fibrillary Acidic Protein, polyclonal antibody (GFAP, 1:500) (Dako, Santa Clara, CA, USA). The secondary antibodies were donkey anti-mouse AlexaFluor 488 conjugated antibody (1:2000) and goat anti-rabbit AlexaFluor 546 conjugated antibody (1:500). Nuclei were stained using 0.1 μM Hoechst for 10 min at room temperature and washed with PBS twice. Ten random images were acquired using an Olympus inverted microscope (IX81, Tokyo, Japan) at 20× magnification, 0.45 numerical aperture, and 7.2 mm working distance, at an excitation: 472/30 nm/emission: 520/35 nm, camera gain: 20.5, exposure time: 1 s for Alex488, excitation: 562/40 nm/emission: 624/40 nm, camera gain: 20.5, exposure time: 0.5 s for Alex 546, and excitation: 377/50 nm, emission: 477/60 nm, camera gain: 20.5, exposure time: 0.04 s for Hoechst. The image analysis was performed using ImageJ Fiji. CD11b (the marker for microglia- higher expression level indicates microgliosis) and GFAP (the marker for astrocytes- higher expression level indicates astrogliosis) staining area and fluorescent intensity were measured in each field.

### 4.10. Glia Shape Analysis

To quantitatively characterize glial morphology, the 2D area was measured for both microglia and astrocytes. Furthermore, ‘Solidity’ ([area of cell (μm^2^)]/[convex area (μm^2^)]) [64] and ‘Transformation Index’ ([perimeter of cell (μm)]^2^/4π [cell area (μm^2^)]) [65] were calculated to categorize the complexity of ramification status, and the ‘Elliptical Form Factor’ (length/breadth) [66] was used to evaluate the degree of cell extension. A cell with more processes displays a higher ‘Transformation Index’, while a higher ‘Elliptical Form Factor’ indicates an elongated cell shape. Unlike microglia, in vitro cultured astrocytes show a complex pattern of outgrowth, so the ‘Sholl’ analysis was only applied to astrocytes to evaluate branching of the outgrowth arbors [67,68]. In the ‘Sholl’ analysis, the number of branches (represented as number of intersections of outgrowth with circled samplings) at different distance from the soma was evaluated (Figure 5G). Image analysis was performed using ImageJ Fiji.

### 4.11. Calcium Imaging of Neonatal Rat MGC

The culture method for MGC and the stimulation with DRG-conditioned medium was described in the former sections. Spines from three rat donors were combined for cell dissociation in each independent replicate. Three independent replicates were performed, and for each independent replicate, 2–3 wells of technical replicates per group were included. The calcium imaging method was mostly the same as DRG cell calcium imaging. The duration of the imaging was set to 700 s. Spontaneous calcium oscillations were evaluated from 0 to 500 s, and response to ATP (a neurotransmitter responsive for glial calcium response) was evaluated from 500 to 700 s. At 500 s, 20 μL of 200 μM ATP (Sigma, Saint Louis, MO, USA) was added to the 180 μL Krebs–Ringer solution in each well, leading to a final ATP concentration of 20 μM. Following calcium imaging, cells were directly fixed in 4% paraformaldehyde for 1 h at room temperature and washed with PBS three times for immunofluorescent staining. A similar methodology was used to evaluate the immunofluorescence as described in Section 4.6 and Section 4.9 with the purpose of differentiating microglia and astrocytes from other types of cells in the calcium imaging.

The ROIs for the calcium imaging analysis were defined manually by comparing stacked images of calcium imaging to the immuno-stained image of the same field. In detail, the images from 0 to 500 s in the calcium imaging were stacked using the ‘Z project’ function of ImageJ Fiji with the projection type set as ‘average intensity’. The ‘sync windows’ function within ImageJ Fiji was used to compare the stacked image to immunofluorescence image. Calcium fluorescent change within the microglial ROIs and astrocyte ROIs were normalized and analyzed using the same derivative method as the analysis in DRG cell calcium imaging. Peak frequency and peak height were calculated for the spontaneous response of microglia and astrocytes, and calcium flux duration and peak height were used to evaluate the calcium response to ATP. To exclude dead cells, ROIs without an ATP-stimulated response were excluded from the analysis. The ATP-stimulated response was defined as peak height from 500 to 700 s higher than 95% of maximum peaks from all ROIs. Calcium flux duration instead of peak frequency was evaluated for ATP-stimulated response because big peaks tend to convolve after ATP stimulation. Processing of the curve analysis was performed using ‘R’.

### 4.12. Bovine Primary DRG Cell Culture

Bovine DRG were dissected from cervical spines of 10–12-month-old cows in an abattoir (3 donors were included in this study). Eight to 10 DRG tissues were dissected and stored in ice-cold modified Krebs–Ringer solution (sucrose 238 mM, KCl 2.5 mM, NaH_2_PO_4_ 1.0 mM, MgCl_2_ 3.8 mM, HEPES 20 mM, and D-glucose 11.0 mM dissolved in ion-free water and filtered with 0.22 μm filter). Then, the DRG were digested with 4 mL of 4 mg/mL collagenase P (Roche, Sigma, Darmstadt, Germany) at 37 °C for 3 h on an orbital shaker. The mechanical dissociation was performed by trituration of the loosened DRG with a pipette tip (P1000) until the solution became turbid. The enzyme was diluted by adding 10 mL DMEM/F12, and the cell suspension was filtered through a 100 μm cell strainer (Falcon, Corning, NY, USA). The cells were carefully pipetted above 15% BSA in PBS and pelleted down by centrifuging at 1800 rpm for 10 min to get rid of the debris. The DRG cells were re-suspended in 0.5 mL of DMEM/F12 (50% *v*/*v*, DMEM from Gibco, Paisley, UK and F-12 Ham from Sigma, Buchs, Switzerland) with 10% FCS (Corning, Woodland, CA, USA), 1% ITS universal culture supplement (Corning, Sigma, Bedford, MA, USA), 1% penicillin/streptomycin, and 2% HEPES (Thermo Fisher, Bleiswijk, The Netherlands). Cells were plated at a density of 5000 neurons/cm^2^ in 16-well chambers (Ibidi, Gräfelfing, Germany) (in 50 μL medium per well) coated with poly-D-lysine and laminin (Sigma-Aldrich, Buchs, Switzerland); coating was performed by incubating with 100 μg/mL PDL for 1 h at 37 °C, washing with PBS three times, and leaving with 2 μg/mL laminin until use. The cells were cultured at 37 °C and 5% CO_2_ for two days before being treated with pro-inflammatory cytokines.

### 4.13. Bovine DRG Cell Cytokine Treatmemt and Calcium Imaging

Bovine recombinant IL-1β, IL-6, and TNF-α (all purchased from R&D, Minneapolis, MN, USA) were all dissolved at a concentration of 10 ng/mL in DMEM/F12 supplemented with 2% HEPES. The method of DRG cytokine treatment was the same as that in the neonatal rat experiment. The treatment lasted for 12 h. Then, a medium change with DMEM/F12 supplemented with 2% HEPES was performed to get rid of cytokines manually added. After another 12 h, the second conditioned media from both cytokine-treated DRG cells and non-treated cells were collected and preserved in −20 °C until used to stimulate spinal cord microglia. The bovine DRG cells were loaded with Fluo-4 using the same method as neonatal rat DRG cells. Calcium imaging was performed using LSM800 confocal (Zeiss, Jena DE) at 10× magnification, 85 μm pinhole, excitation of 488 nm, and an emission at 509 nm. The image acquisition rate was set at one image per second, and the recording time was set to 110 s. The first 100 s were taken to study spontaneous calcium response. To identify neurons from other types of cells, at the 100th second, 5 μL of 500 mM potassium chloride (final concentration at 50 mM) was added to the DRG cells, and recording was continued for another 10 s. Cells with an immediate calcium response to potassium were recognized as neurons. The calcium imaging data analysis was the same as described for calcium imaging of neonatal rat DRG cells.

### 4.14. Bovine DRG Cell Type Characterization Using Immunofluorescence

After calcium imaging, DRG cells were immediately fixed in 4% formalin for 1 h at room temperature. Different cell types in the bovine DRG cell culture were characterized using immunofluorescence. Five wells were stained with primary antibodies including the following: mouse anti-TUBB3 (1:200) to stain neurons; rabbit anti-GFAP (1:500) to stain satellite cells; and goat anti-Iba1 (ionized calcium binding adaptor molecule 1, 1:200, Abcam, Amsterdam, The Netherlands) to stain macrophages. Another 5 wells were stained with primary antibodies, including the following: mouse anti-TUBB3 (1:200) to stain neurons; rabbit anti-GFAP (1:500) to stain satellite cells; and goat anti-SOX10 (SRY-related HMG-box 10, 1:50, St John’s Laboratory Ltd., London, UK) to stain Schwann cells. The DRG cells were incubated with primary antibodies overnight at 4 °C. After 3 washes of PBS-T, cells were incubated with secondary antibodies including donkey anti-goat AlexaFluor 488 conjugated antibody (1:500, Thermo Fisher, Eugene, OR, USA) and donkey anti-rabbit AlexaFluor 680 (1:500, Thermo Fisher, Eugene, OR, USA) conjugated antibody for 1 h at room temperature. After another wash with PBS-T, cells were incubated with goat anti-mouse AlexaFluor 555 conjugated antibody (1:500, Thermo Fisher, Eugene, OR, USA) for 1 h at room temperature. Then, the cells were washed with PBS-T 3 times. Nuclei were stained using 1 μM Hoechst for 10 min at room temperature and washed with PBS twice. Images were taken using LSM800 confocal at 40× magnification, 80 μm Pinhole, at an excitation: 679 nm/emission: 702 nm for Alex680, excitation: 553 nm/emission: 568 nm for Alex 555, excitation: 493 nm/emission: 517 nm for Alex 488, and excitation: 348 nm, emission: 455 nm for Hoechst.

### 4.15. Bovine Spinal Cord Microglia Culture

Bovine spinal cords were dissected from cervical spines of 10–12-month-old cows in abattoir (3 donors were included in this study). The spinal cords were stored and processed in ice-cold modified Krebs–Ringer solution (sucrose 238 mM, KCl 2.5 mM, NaH_2_PO4 1.0 mM, MgCl_2_ 3.8 mM, HEPES 20 mM, and D-glucose 11.0 mM). Meninges tissues including dura matter and pia matter were carefully removed (Figure 9A,B). The spinal cords were minced and digested in 6 mL of 4 mg/mL collagenase P (Roche, Sigma, Mannheim, Germany) at 37 °C for 3 h on an orbital shaker. Mechanical trituration was performed by passing the tissue through 18 G (twice) and 21 G (twice) needles sequentially. The dissociated tissue was filtered through a 70 μm cell strainer. The cells were carefully pipetted above 15% BSA in PBS and pelleted down by centrifuging at 1800 rpm for 10 min to reduce debris. Cells were resuspended in 0.4 mL DMEM/F12 with 10% FCS, 1% ITS, 1% penicillin/streptomycin, and 2% HEPES. The cells were plated at a density of 10,000 cells /cm^2^ in 16-well chambers coated with poly-D-lysine and cultured at 37 °C and 5% CO_2_. To attain a good attachment and confluency, the cells were maintained for at least 5 days before stimulated by the DRG-conditioned media.

### 4.16. Immunofluorescence and Calcium Imaging of Bovine Microglia Stimulated by the Second DRG-Conditioned Medium

Once the spinal cord microglia reached a confluency of around 50%, the culture media were changed to the second conditioned media collected from bDRG cells. After 48 h of DRG-conditioned medium stimulation, calcium imaging of the spinal cord microglia was performed as described for calcium imaging of neonatal rat MGC. To perform the calcium imaging, LSM800 confocal was set as 10× magnification, 85 μm pinhole, excitation of 488 nm, and emission at 509 nm. The image acquisition rate was set at one image per second, and the recording time was set to 200 s. The first 100 s were taken to study spontaneous calcium response. At 100 s, 5 μL of 200 μM ATP were added to the microglia in 45 μL Krebs–Ringer solution for a final ATP concentration of 20 μM. Another 100 s were recorded to study the microglial response to ATP.

After calcium imaging, the microglia were immediately fixed in 4% formalin for 1 h and then washed in PBS 3 times. Immunofluorescence was performed using the same method described in former sections. Primary antibodies were goat anti-Iba1 antibody (1:200, Abcam, Amsterdam, The Netherlands) and rabbit anti-GFAP antibody (1:500, Dako, Santa Clara, CA, USA). Secondary antibodies were donkey anti-goat AlexaFluor 488 conjugated antibody and donkey anti-rabbit AlexaFluor 680.

In the bovine spinal cord cell culture, only Iba1-stained microglia were detected, while GFAP-stained astrocytes could not be found. The calcium imaging analysis of bovine microglia was the same as that of neonatal rat MGC. Likewise, the ROIs for the calcium imaging analysis were defined manually by comparing stacked image of calcium imaging to the immuno-stained image of the same field. Iba1 expression level was also analyzed based on the same method in CD11b expression level analysis in neonatal rat MGC.

### 4.17. Statistics

The statistics was first performed based on technical replicates (different image fields for immunofluorescence or different cells for calcium imaging). For DRG cell calcium imaging and the CGRP immuno-staining normalized by TUBB3, data were not normally distributed (Shapiro–Wilk normality test *p* < 0.1), so the Mann–Whitney test was used to statistically compare the two groups. For the proportion of neurons expressing CGRP in each field, since data were normally distributed (Shapiro–Wilk normality test *p* > 0.1), a two-sampled t-test was used for the statistical analysis. For the comparison among multiple groups in MGC immunofluorescence, shape, and calcium imaging analysis, the data were not all normally distributed, so the Wilcoxon rank-sum test was performed for the statistical analysis. P-values lower than 0.05 were considered significant.

Then, statistical analysis was performed using data points of the median level of each independent experiment using different donors. A paired *t*-test was performed to compare between groups. *p*-values lower than 0.05 were considered significant.

## 5. Conclusions

In summary, the cultured DRG neurons stimulated by the inflammatory cytokines TNF-α, IL-1β, and IL-6 can activate the glia isolated from spinal cord. This in vitro communication between DRG and spinal cord glia was observed in cells from both neonatal rat and adult cattle. The in vitro model may provide an easy alternative to animal models of pain in line with 3R principles and can be used in the pilot study of molecular mechanism and in the discovery of new drugs treating chronic pain transition.

## Figures and Tables

**Figure 1 ijms-22-09725-f001:**
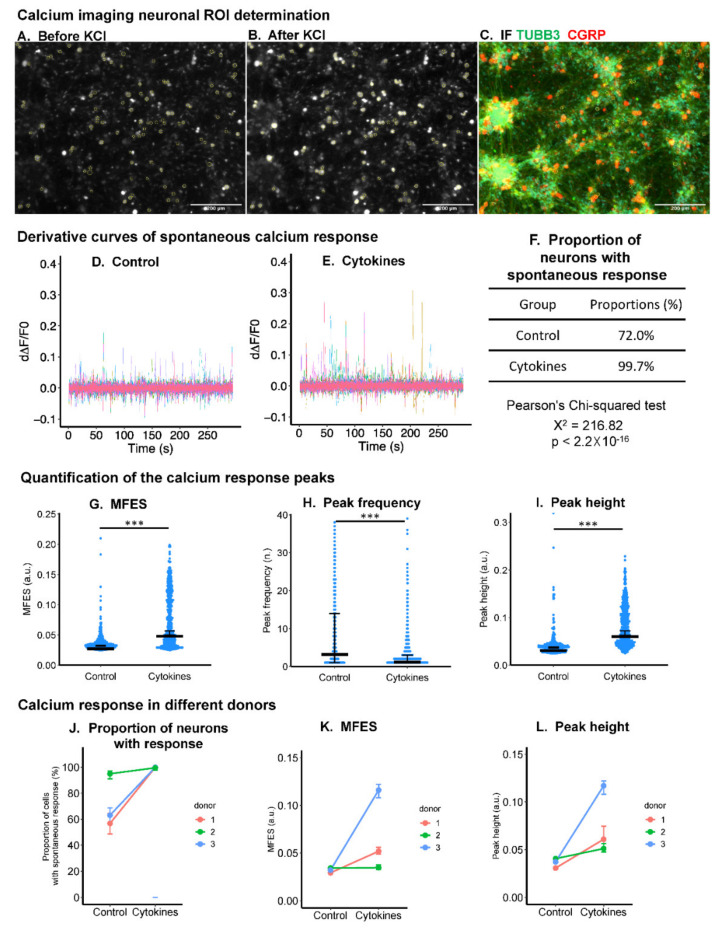
Calcium imaging (Fluo-4) analysis of primary neonatal rat DRG culture treated with (‘Cytokine’ group) and without (‘Control’ group) TNF-α, IL-1β, and IL-6 (each 10 ng/mL). (**A**,**B**) A response to KCl (50 mM) was used to select the region of interest (ROI) representing viable neurons. (**C**) KCl distinguished neurons were validated using immunostaining with neuronal marker tubulin β-III (TUBB3). Scale bars equal 200 μm. (**D**,**E**) Derivative curves of the change in fluorescence (intracellular calcium) over time showed a higher spontaneous calcium activity in the ‘Cytokine’ group compared with the ‘Control’ group. (**F**–**I**) Cytokine treatment enhanced the ‘Proportion of neurons with response’, ‘Maximum fluorescence elevation speed (MFES)’, and ‘Peak height’, but it decreased the ‘Peak frequency’. (**J**–**L**) The ‘Cytokine’ group showed higher ‘Proportion of neurons with response’ and ‘Peak height’ than the ‘Control’ group in all of the three independent experiments using different donors. For (**F**–**I**), a Chi-square test was used to statistically analyze ‘Proportion of neurons with response’; the Mann–Whitney test was used to analyze ‘MFES’, ‘Peak frequency’, and ‘Peak height’. *** *p* < 0.001, *n* = 148 and 150 cells for the ‘Control’ group and ‘Cytokine’ group, respectively. For (**J**–**L**), a paired *t*-test did not find a significant difference between groups in terms of median levels of each donor, *n* = 3 donors. In these plots, middle bars display median levels; error bars represent 95% confident intervals.

**Figure 2 ijms-22-09725-f002:**
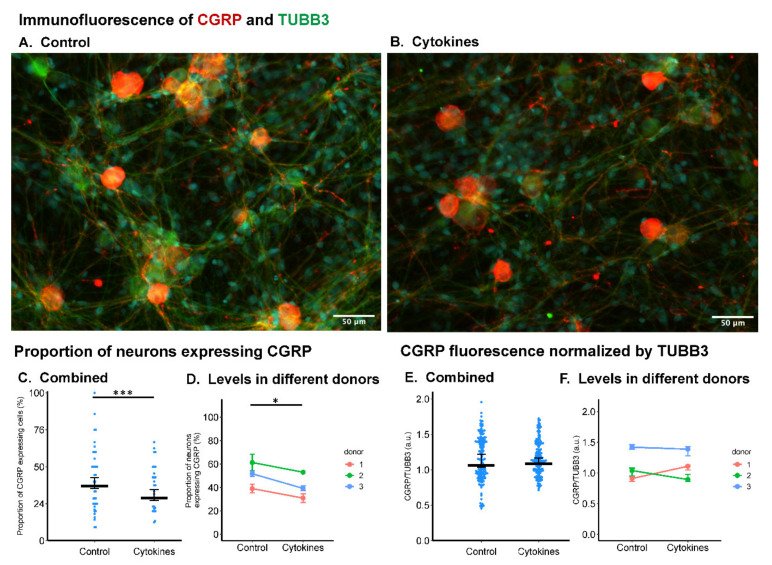
Immunofluorescent analysis of calcitonin gene-related peptide (CGRP) expression in primary neonatal rat DRG culture treated with (‘Cytokine’ group) or without (‘Control’ group) TNF-α, IL-1β, and IL-6 (each 10 ng/mL). (**A,B**) Representative images of the immunostaining. CGRP was stained in red and the neuronal marker TUBB3 was stained in green. Scale bars equal 50 μm. (**C,D**) The proportion of neurons expressing CGRP was decreased by the cytokine treatment compared with control, and different donors were showing the same trend. *** *p* < 0.001 by two-sampled *t*-test. *N* = 40 and 30 fields for the ‘Control’ group and ‘Cytokine’ group, respectively. Middle bars represent mean level; error bars represent standard error. (**E,F**) CGRP fluorescent intensity normalized by the level of TUBB3 showed no significant difference between groups. For D and F, the Mann–Whitney test was used for the statistical analysis since the data were not normally distributed. *N* = 40 and 30 fields for the ‘Control’ group and ‘Cytokine’ group, respectively. For (**D**,**F**), a paired *t*-test was performed. * *p* < 0.05, *n* = 3 donors per group. Middle bars display median levels, and error bars represent 95% confident intervals.

**Figure 3 ijms-22-09725-f003:**
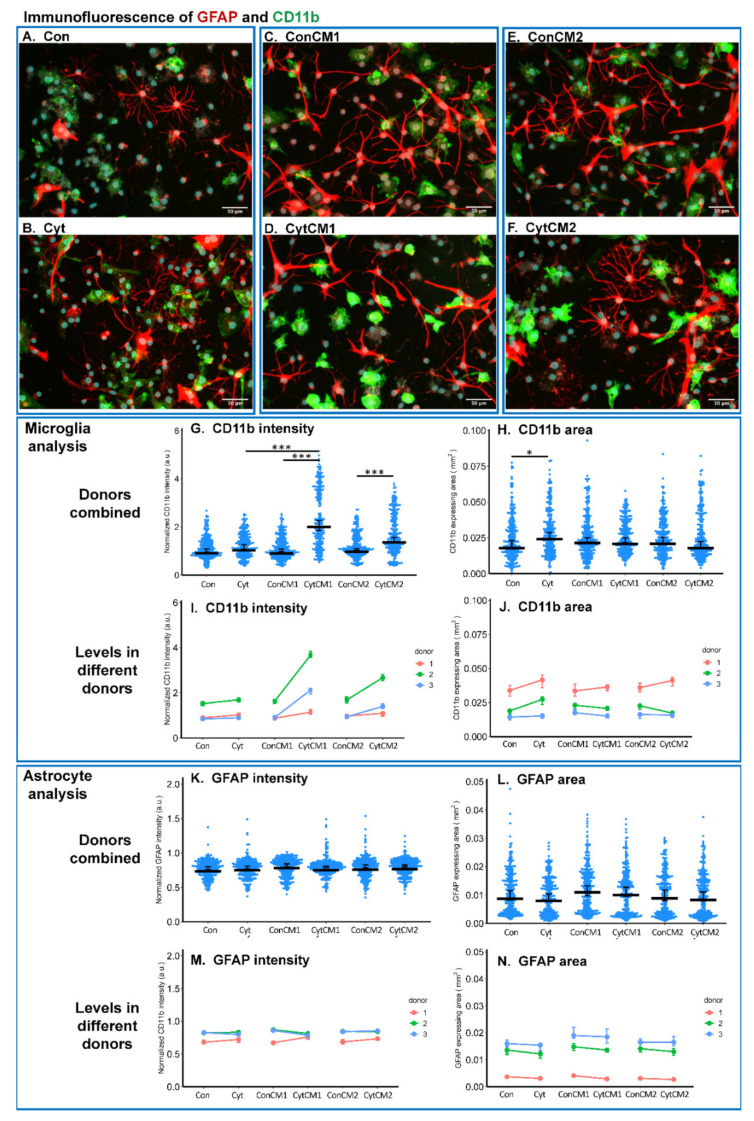
Immunofluorescent analysis of CD11b (marker to identify microglia and expression level associated with microgliosis) expression in microglia and glial fibrillary acidic protein (GFAP) (marker to identity astrocytes and expression level associated with astrogliosis) expression in astrocytes in the neonatal rat mixed glial culture (MGC). ‘Cyt’ and ‘Con’ represent the MGCs treated with or without TNF-α, IL-1β, and IL-6 (each 10 ng/mL), respectively; ‘CytCM1’ and ‘CytCM2’ represent the MGCs treated with either the first or the second DRG-conditioned media, in which the DRG cells were pre-treated with cytokines. The second DRG-conditioned medium was collected after a medium change and thus had no inflammatory cytokines initially added. ‘ConCM1’ and ‘ConCM2’ represent MGCs treated with the first and second DRG cell-conditioned media, in which the DRG cells were not pre-treated with inflammatory cytokines. (**A**–**F**) Representative images of the immunostaining. CD11b was stained in green and GFAP was stained in red. Scale bars equal 50 μm. (**G**) Both the first and second collected cytokine-treated DRG-conditioned media (‘CytCM1’ and ‘CytCM2’) significantly upregulated CD11b expression compared with the corresponding non-cytokine-treated DRG-conditioned media (‘ConCM1’ and ‘ConCM2’). (**H**) Applying inflammatory cytokines to the MGC (‘Cyt’) increased the confluency of CD11b-expressing microglia as compared with the non-treated control (‘Con’). (**I,J**) The findings in (**G**,**H**) are showing the same trend among the three different donors. For (**G**–**J**), * *p* < 0.05 and *** *p* < 0.001 by Wilcoxon rank-sum test. *N* = 246–266 fields per group. (**K**–**N**) The expression level of GFAP and confluency of astrocytes did not show significant differences among groups. For (**G**,**H**,**K**,**L**), the Wilcoxon rank-sum test was used for the statistical analysis, *n* = 246–266 fields per group. For (**I**,**J**,**M**), and N, in each group, the median level of each donor was first averaged; then, a paired *t*-test was used to compare the median data points between groups. No significant difference could be observed, *n* = 3 donors. For these plots, middle bars display median levels, and error bars represent 95% confident intervals.

**Figure 4 ijms-22-09725-f004:**
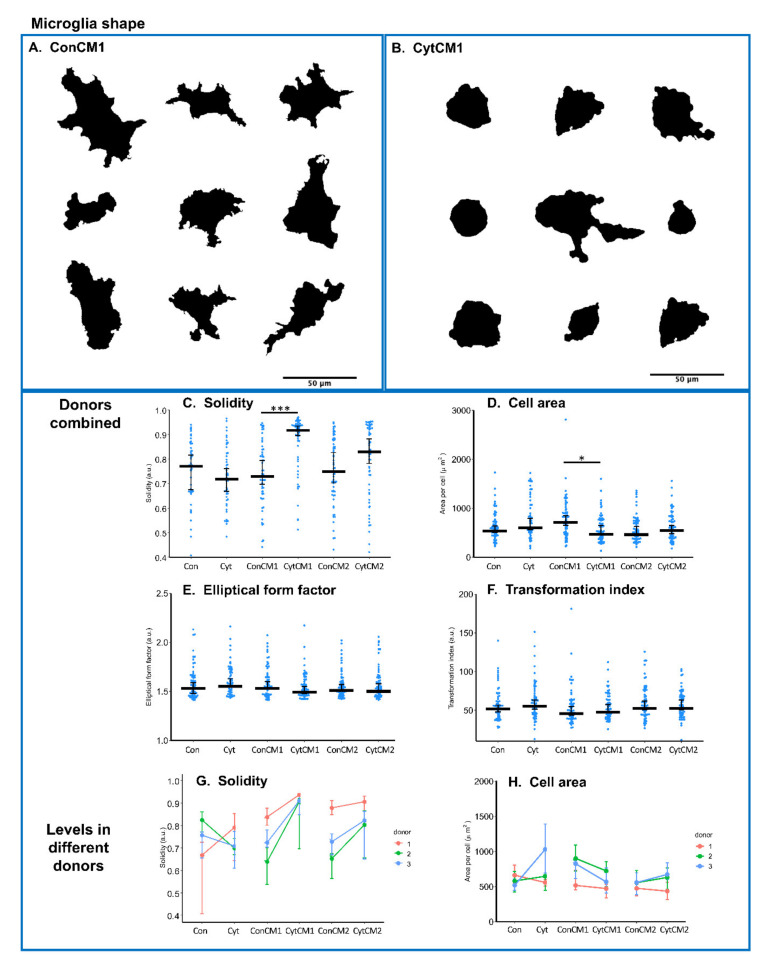
Shape analysis of microglia. ’Cyt’ and ’Con’ represent the MGCs treated with or without TNF-α, IL-1β, and IL-6 (each 10 ng/mL), respectively; ’CytCM1’ and ’CytCM2’ represent the MGCs treated with either the first or the second collected DRG-conditioned media, in which the DRG cells were pre-conditioned with cytokines. The second DRG-conditioned media were collected after a medium change and thus had no inflammatory cytokines initially added. ‘ConCM1’ and ‘ConCM2’ represent MGCs treated with the first and second DRG cell-conditioned media, in which the DRG cells were not pre-conditioned with inflammatory cytokines. (**A**,**B**) Representative images from the ‘ConCM1’ and ‘CytCM1’ groups showing shape change. Scale bars equal 50 μm. (**C**) Cell solidity was significantly larger in the ‘CytCM1’ group than in the ‘ConCM1’ group. (**D**) Cell area of microglia in the ‘CytCM1’ group was significantly smaller compared with the ‘ConCM1’ group. (**E**,**F**) The ‘Elliptical form parameter’ and ‘Transformation index’ showed no significant difference among groups. (**G**,**H**) The increase in cell solidity and decrease in cell area in the ‘CytCM1’ group as compared with the ‘ConCM1’ group were observed in all three donors. For (**C**–**F**), * *p* < 0.05 and *** *p* < 0.001 by Wilcoxon rank-sum test, *n* = 60–61 cells per group. For (**G**,**H**), in each group, the median level of each donor was first calculated; then, a paired *t*-test was used to compare these median data points between groups. No significant difference could be observed, *n* = 3 donors. For plots (**C**–**H**), middle bars display median levels, and error bars represent 95% confident intervals.

**Figure 5 ijms-22-09725-f005:**
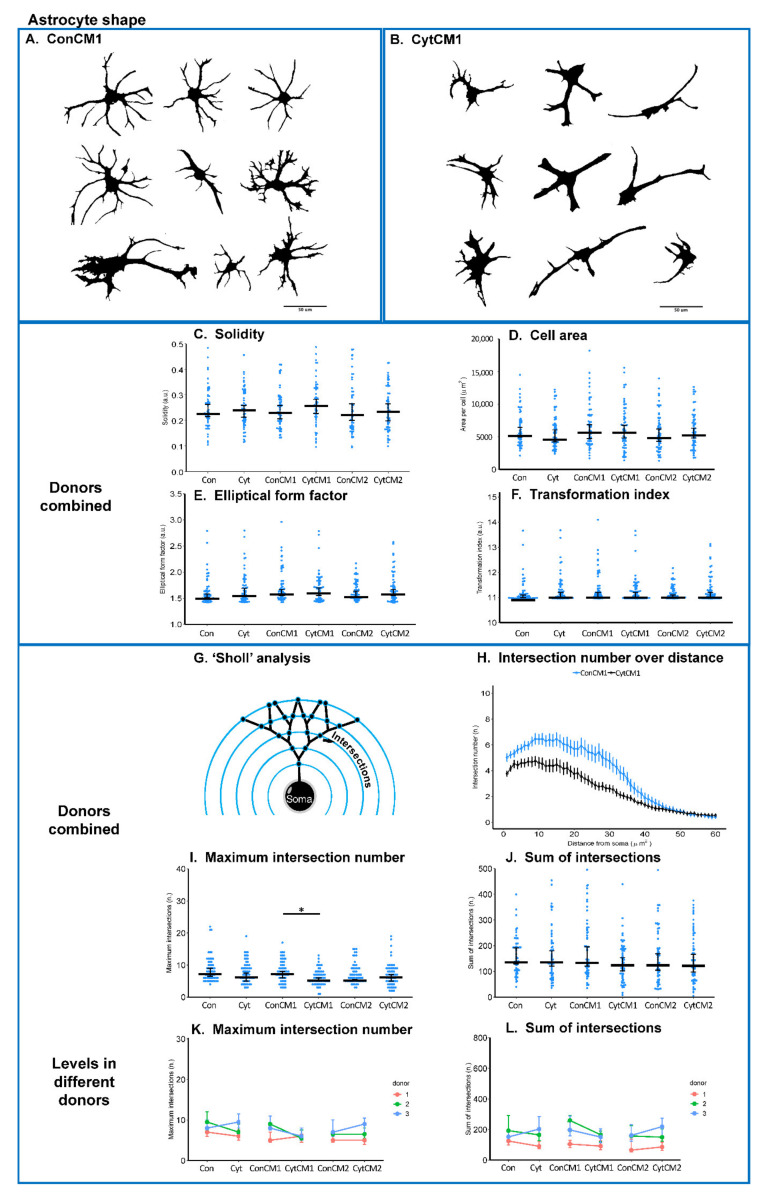
Shape analysis of astrocytes. ‘Cyt’ and ‘Con’ represent the MGCs treated with or without TNF-α, IL-1β, and IL-6 (each 10 ng/mL), respectively; ‘CytCM1’ and ‘CytCM2’ represent the MGCs treated with either the first or the second collected DRG-conditioned media, in which the DRG cells were pre-treated with cytokines. ‘ConCM1’ and ‘ConCM2’ represent MGCs treated with the first and second DRG cell-conditioned media, in which the DRG cells were not pre-treated with inflammatory cytokines. (**A**,**B**) Representative images from the ‘ConCM1’ and ‘CytCM1’ groups showing shape change. Scale bars equal 50 μm. (**C**–**F**) The ‘Cell solidity’, ‘Cell area’, ‘Elliptical form parameter’, and ‘Transformation index’ showed no significant difference among groups. (**G**) Schematic showing the ‘Sholl’ analysis of the cellular process pattern. Sampling circles intersect with the astrocyte process arbors providing information on the number of branches (represented as number of intersections) at different distance from the soma. (**H**) Number and distribution of intersections at different distances from the soma. Note that the curve representing the ‘CytCM1’ group (black line) is mostly below the ‘ConCM1’ group (blue line). (**I**) ‘Maximum intersection number’ in the ‘CytCM1’ group was significantly larger than in the ‘ConCM1’ group. (**J**) ‘Sum of intersections’ did not show any significant difference among groups. (**K**) Compared with the ‘ConCM1’ group, the ‘CytCM1’ group showed a decreased ‘Maximum intersection number’ in two out of the three independent replicates (**L**) and a lower ‘Sum of intersections’ in all the three independent replicates. For (**C**–**F**,**I**,**J**), * *p* < 0.05 by the Wilcoxon rank-sum test. *N* = 57–61 cells per group. For (**K**,**L**), in each group, the median level of each donor was first calculated; then, a paired *t*-test was used to compare these median data points between groups. No significant difference could be observed, *n* = 3 donors. For these plots, middle bars represent median levels, and error bars represent 95% confident intervals.

**Figure 6 ijms-22-09725-f006:**
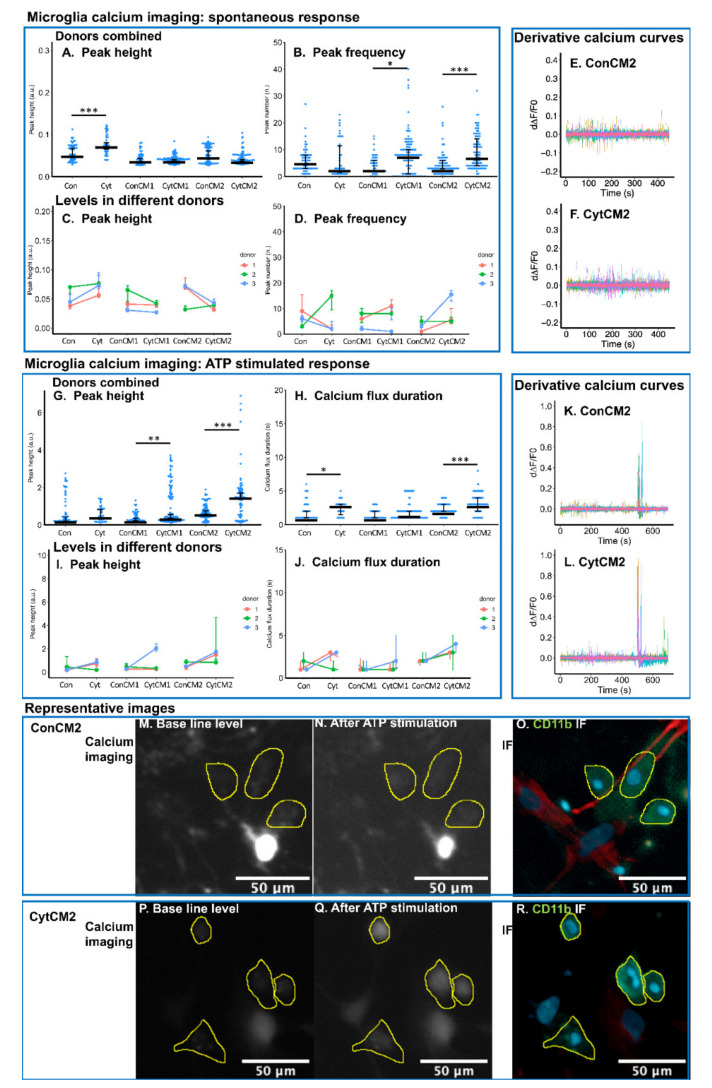
Calcium imaging analysis of microglia. ‘Cyt’ and ‘Con’ represent the MGCs treated with or without TNF-α, IL-1β, and IL-6 (each 10 ng/mL), respectively; ‘CytCM1’ and ‘CytCM2’ represent the MGCs treated with either the first or the second collected DRG-conditioned media, in which the DRG cells were pre-treated with cytokines. ‘ConCM1’ and ‘ConCM2’ represent MGCs treated with the first and second DRG cell-conditioned media, in which the DRG cells were not pre-treated with inflammatory cytokines. (**A**) Spontaneous calcium peak height in microglia was elevated by directly treating the MGC with the inflammatory cytokines (‘Cyt’ group) as compared with the non-treated control (‘Con’ group). (**B**) Spontaneous calcium peak frequency was enhanced by the first and second conditioned media from cytokine-treated DRG (‘CytCM1’ and ‘CytCM2’) compared with the corresponding conditioned media from non-treated DRG (‘ConCM1’ and ‘ConCM2’), respectively. (**C**) Spontaneous calcium peak height in all three donors was increased in the ‘Cyt’ group as compared with the ‘Con’ group. (**D**) Spontaneous calcium peak frequency in two out of three donors was increased in the ‘CytCM2’ group as compared with the ‘ConCM2’ group. (**E,F**) Derivative curves of calcium fluorescent change over time. (**G**) ATP-stimulated calcium peak height was elevated by both the first and second conditioned media from cytokine-treated DRG (‘CytCM1’ and ‘CytCM2’) compared with corresponding conditioned media from non-cytokine-treated DRG (‘ConCM1’ and ‘ConCM2’), respectively. (**H**) ATP-stimulated calcium flux duration was higher in the ‘Cyt’ group than in the ‘Con’ group and was also higher in the ‘CytCM2’ group than in the ‘ConCM2’ group. (**I**) The elevated ATP-stimulated calcium peak height was observed in two out of three donors comparing ‘CytCM2’ with ‘ConCM2’. (**J**) The increased ATP-stimulated calcium flux duration was observed in all three donors in ‘CytCM2’ compared with ‘ConCM2’. (**K,L**) Derivative curves of calcium fluorescent change over time showed a higher microglial calcium response to ATP in the ‘CytCM2’ group as compared with the ‘ConCM2’ group. (**M**–**R**) Representative images showing a more intensive ATP-stimulated response in microglia in the ‘CytCM2’ group ((**Q**) versus (**P**)) as compared with the ‘ConCM2’ group ((**N**) versus (**M**)). (**O**–**R**) show the method of using immunofluorescent staining to distinguish microglia from other types of cells. Scale bars equal 50 μm. For (**A**,**B**,**G**,**H**), * *p* < 0.05, ** *p* < 0.01, and *** *p* < 0.001 by the Wilcoxon rank-sum test. *N* = 44–119 cells per group. For (**C**,**D**,**I**,**J**), in each group, the median level of each donor was first calculated; then, a paired *t*-test was used to compare these median data points between groups. No significant difference could be observed, *n* = 3 donors. For these plots, middle bars display median levels, and error bars represent 95% confident intervals.

**Figure 7 ijms-22-09725-f007:**
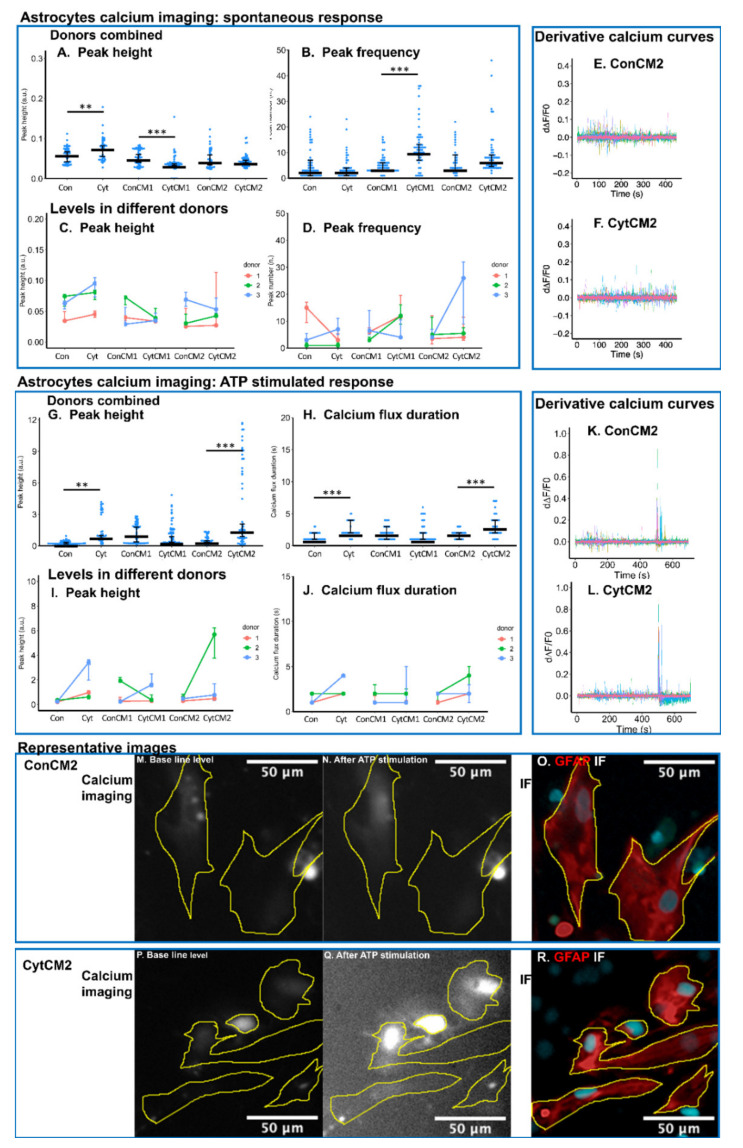
Calcium imaging analysis of astrocytes. ‘Cyt’ and ‘Con’ represent the MGCs treated with or without TNF-α, IL-1β, and IL-6 (each 10 ng/mL), respectively; ‘CytCM1’ and ‘CytCM2’ represent the MGCs treated with either the first or the second collected DRG-conditioned media, in which the DRG cells were pre-conditioned with cytokines. ‘ConCM1’ and ‘ConCM2’ represent MGCs treated with the first and second DRG cell-conditioned media, in which the DRG cells were not pre-treated with inflammatory cytokines. (**A**) Spontaneous calcium peak height in astrocytes was increased by directly treating the MGC with the inflammatory cytokines (‘Cyt’ group) as compared with the non-treated control (‘Con’ group). (**B**) Spontaneous calcium peak frequency was elevated by the first conditioned medium from cytokine-treated DRG (‘CytCM1’) compared with the first conditioned medium from non-cytokine-treated DRG (‘ConCM1’). (**C**) Spontaneous calcium peak height in all three donors was increased in the ‘Cyt’ group compared with the ‘Con’ group. (**D**) Spontaneous calcium peak frequency in two out of three independent experiments was increased in the ‘CytCM1’ group compared with the ‘ConCM1’ group. (**E,F**) Derivative curves of calcium fluorescent change over time. (**G**,**H**) ATP-stimulated calcium peak height and calcium flux duration were more elevated in the ‘Cyt’ group than in the ‘Con’ group, and they were also higher in the ‘CytCM2’ group than in the ‘ConCM2’ group. (**I**) The elevated ATP-stimulated calcium peak height was observed in all three donors comparing ‘Cyt’ with ‘Con’ and comparing ‘CytCM2’ with ‘ConCM2’. (**J**) The increased ATP-stimulated calcium flux duration was observed in two out of three donors comparing ‘Cyt’ with ‘Con’ and comparing ‘CytCM2’ with ‘ConCM2’. (**K**,**L**) Derivative curves of calcium fluorescent change over time showed a higher astrocytic calcium response to ATP in the ‘CytCM2’ group as compared with the ‘ConCM2’ group. (**M**–**R**) Representative images showing a more intensive ATP-stimulated response in astrocytes in the ‘CytCM2’ group ((**Q**) versus (**P**)) as compared with the ‘ConCM2’ group ((**N**) versus (**M**)). (**O**,**R**) show the method of using immunofluorescent staining to distinguish the ROI of astrocytes from other types of cells. Scale bars equal 50 μm. For (**A**,**B**,**G**,**H**), ** *p* < 0.01, and *** *p* < 0.001 by the Wilcoxon rank-sum test. *N* = 40–65 cells per group. For (**C**,**D**,**I**,**J**), in each group, the median level of each donor was first calculated; then, a paired *t*-test was used to compare the median data points between groups. No significant difference could be observed, *n* = 3 donors. For these plots, middle bars display median levels, and error bars represent 95% confident intervals.

**Figure 8 ijms-22-09725-f008:**
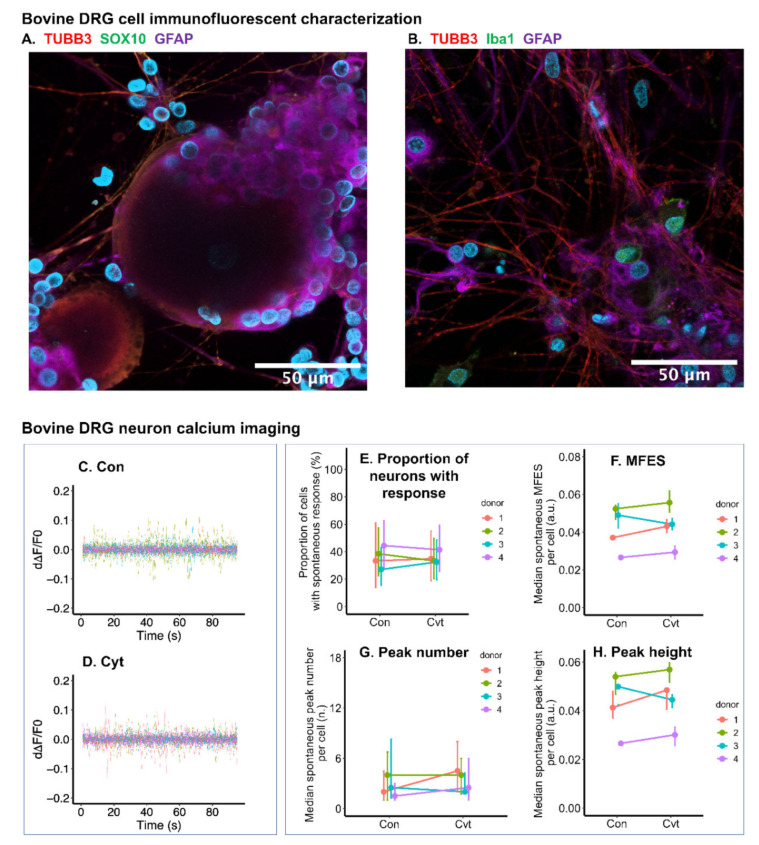
Bovine DRG primary cell (bDRG) immunofluorescent characterization and calcium imaging. ‘Cyt’ and ‘Con’ represent the bDRG treated with or without TNF-α, IL-1β, and IL-6 (each 10 ng/mL), respectively. (**A**) Many GFAP-labeled satellite cells were observed around TUBB3-labeled neurons. (**B**) A few macrophages were observed by immunostaining of Iba1. (**C,D**) Derivative curves of calcium fluorescent change over time. (**E**–**H**) Proportion of neurons with spontaneous response, maximum fluorescent elevation speed (MFES), peak number, and peak height were not found to be significantly different comparing ‘Cyt’ with ‘Con’ using a paired *t*-test (*n* = 4 donors). Scale bars equal 50 μm. For (**E**–**H**), error bars represent 95% confident intervals.

**Figure 9 ijms-22-09725-f009:**
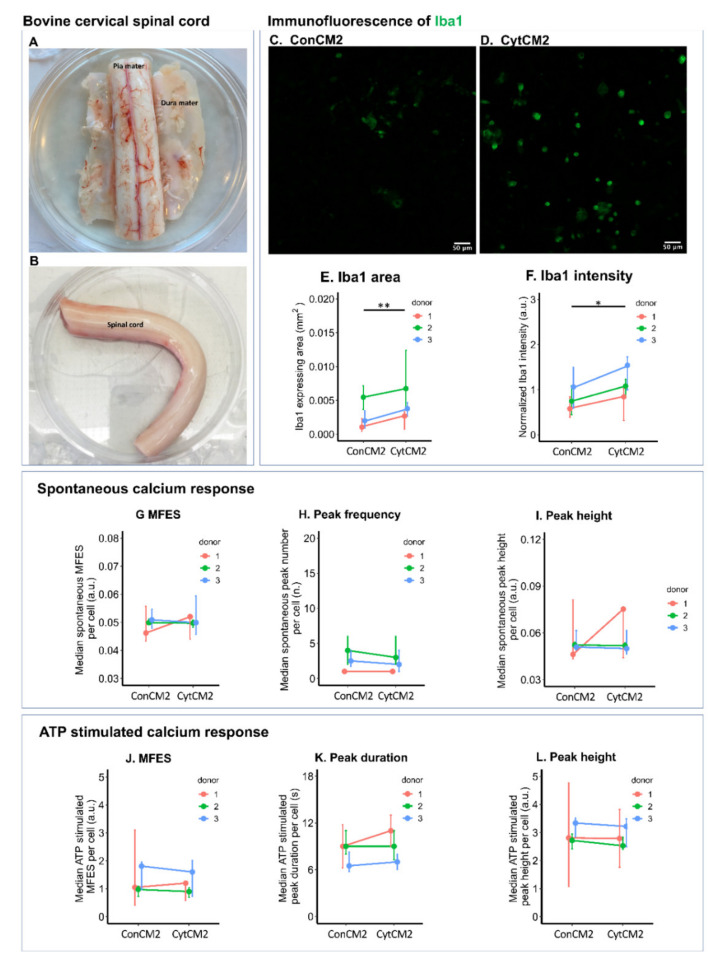
Iba1 expression and calcium imaging of bovine spinal cord microglia stimulated by bovine DRG cell-conditioned medium. ‘CytCM2’ represents the microglia treated with the secondly collected DRG-conditioned media, in which the DRG cells were pre-conditioned with cytokines. ‘ConCM2’ represents the microglia treated with the second DRG-conditioned medium, in which the DRG cells were not pre-treated with inflammatory cytokines. (**A**,**B**) The cervical spinal cord obtained from abattoir. (**C**,**D**) Representative immunofluorescent images showing a higher Iba1 expression for the ‘CytCM2’ group compared with the ‘ConCM2’ group. Scale bars equal 50 μm. I Iba1-expressing area was significantly increased in all three donors for the ‘CytCM2’ group compared with the ‘ConCM2’ group. ** *p* < 0.01 by paired *t*-test, *n* = 3 donors. (**E**,**F**) Iba1-expressing level was significantly increased in all three donors for the ‘CytCM2’ group compared with the ‘ConCM2’ group. * *p* < 0.05 by paired *t*-test, *n* = 3 donors. (**G**–**I**) In spontaneous calcium response of microglia, maximum fluorescent elevation speed, peak number, and peak height were not significantly different comparing ‘ConCM2’ with ‘CytCM2’ (paired *t*-test, *n* = 3 donors). (**J**–**L**) In ATP-stimulated calcium response of microglia, the maximum fluorescent elevation speed, peak duration, and peak height were not significantly different comparing ‘ConCM2’ with ‘CytCM2’ (paired *t*-test, *n* = 3 donors). For these plots, middle bars display median levels, and error bars represent 95% confident intervals.

## Data Availability

Data that supports the central findings is contained within the article.

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
