# Peer review of "In Vitro Model to Investigate Communication between Dorsal Root Ganglion and Spinal Cord Glia"

_ijms, 2021, doi:10.3390/ijms22189725_

Round 1
Reviewer 1 Report
The authors have adequately addressed most of my concerns and have greatly improved the rigour of their manuscript. The addition of bovine DRG cultures is really interesting and novel. I have a few minor comments for improvements.
- The discussion of the neonatal rat cultures that is currently in the intro could be moved to the discussion.
- In the discussion please add a few sentences summarizing what has been done with bovine DRGs. To my knowledge there is little to no work with them.
- The authors say that the neonatal rat findings were validated in the bovine cultures. In reality, a number of data pieces did not replicate and astrocytes were not examined (why?) in bovine cultures. Please correct these statements
Finally, to my knowledge, there is essentially no work done with bovine DRGs. The novelty of this could be stressed more in the discussion. I think it would be great if the authors have a follow-up study and manuscript thoroughly characterizing their cultures and comparing bovine and human DRG gene expression patterns, as they alluded to. This could be a great resource for the intervertebral disc field that often works with bovine spines as well as other neuroscientists that want to work with DRGs from large mammals but lack access to human and non-human primate DRGs.
Author Response
Response to Reviewer 1 Comments
The authors have adequately addressed most of my concerns and have greatly improved the rigour of their manuscript. The addition of bovine DRG cultures is really interesting and novel. I have a few minor comments for improvements.
- The discussion of the neonatal rat cultures that is currently in the intro could be moved to the discussion.
Respond:
Many thanks for the suggestion. The discussion of neonatal rat cultures has been moved to discussion part in line 505-510.
- In the discussion please add a few sentences summarizing what has been done with bovine DRGs. To my knowledge there is little to no work with them.
Respond:
The summarization of what has been done with bovine DRGs has been added in line 511-515 and has been copied as follows:
“Due to limited access to human DRG tissue, DRG from large animals is a promising alternative. For example, DRG neuronal subtypes were classified in non-human primate based on transcriptome [48]. Unlike macaque, bovine tissue can be easily accessed from abattoir without additional animal euthanasia. The culture of bovine DRG cells has been reported [49] but has not been widely used in the study of pain-associated biology.”
- The authors say that the neonatal rat findings were validated in the bovine cultures. In reality, a number of data pieces did not replicate and astrocytes were not examined (why?) in bovine cultures. Please correct these statements
Respond:
Many thanks for this comment. We also realize that it is not correct to use the word 'validate'. The statements have been changed to “Adult bovine DRG and spinal cord cell cultures were also tested, as an alternative large animal model, and results were compared with the neonatal rat findings.” in line 20-22, 77 and 398.
Only the positive findings from neonatal rat cell culture were evaluated in the bovine cell culture.
Finally, to my knowledge, there is essentially no work done with bovine DRGs. The novelty of this could be stressed more in the discussion. I think it would be great if the authors have a follow-up study and manuscript thoroughly characterizing their cultures and comparing bovine and human DRG gene expression patterns, as they alluded to. This could be a great resource for the intervertebral disc field that often works with bovine spines as well as other neuroscientists that want to work with DRGs from large mammals but lack access to human and non-human primate DRGs.
Respond:
We would like to thank the reviewer for this great suggestion. The prospect has been added in the discussion in line 515-516.
“The gene expression profile of bovine DRG needs to be compared with human in the future to support the use of bovine DRG culture.”

Reviewer 2 Report
The communication between peripheral DRG and the central spinal cord glia may explain why DRG, that are exposed to cytokines released from degenerative, inflamed IVDs, can lead to central glial activation in spinal cord.
This article showed that DRG conditioned medium elevated CD11b expression and calcium signal in neonatal rat microglia and enhanced Iba1 expression in adult bovine microglia. Thus, cytokine treatment induced a DRG-mediated microgliosis. The in vitro model described here, characterizing the DRG-glia communication, may represent an alternative to animal pain models.
Minor:
In the Abstract, “Adult bovine DRG and spinal cord cell cultures were performed to validate the neonatal rat findings.” should be replaced by “Adult bovine DRG and spinal cord cell cultures were tested to ……”
Author Response
The communication between peripheral DRG and the central spinal cord glia may explain why DRG, that are exposed to cytokines released from degenerative, inflamed IVDs, can lead to central glial activation in spinal cord.
This article showed that DRG conditioned medium elevated CD11b expression and calcium signal in neonatal rat microglia and enhanced Iba1 expression in adult bovine microglia. Thus, cytokine treatment induced a DRG-mediated microgliosis. The in vitro model described here, characterizing the DRG-glia communication, may represent an alternative to animal pain models.
Minor:
In the Abstract, “Adult bovine DRG and spinal cord cell cultures were performed to validate the neonatal rat findings.” should be replaced by “Adult bovine DRG and spinal cord cell cultures were tested to ……”
Respond:
Many thanks for the comment. The abstract has been revised accordingly.
“Adult bovine DRG and spinal cord cell cultures were performed to validate the neonatal rat findings.” has been changed to “Adult bovine DRG and spinal cord cell cultures were also tested, as an alternative large animal model, and results were compared with the neonatal rat findings.” in line 18-20.

This manuscript is a resubmission of an earlier submission. The following is a list of the peer review reports and author responses from that submission.
Round 1
Reviewer 1 Report
The authors have carried out a lot of pretty difficult cell isolations and measurements. However, apart from the problem of their acknowledged donor variability (additional experiments required), the design of experiments and levels of cytokines used, seems very far from any conditions seen in vivo. Whether the results have any relevance to hronic pain development, will require much justification.
However, maybe it is of interest to show that (stimulated- overstimulated?) DRGs can under some circumstances influence the glial cell populations, but I don't have any knowledge of this area.
Reviewer 2 Report
The manuscript by Ma et al. uses two different in vitro culture systems to investigate how soluble factors released by cells in dissociated dorsal root ganglia cultures can affect spinal microglia and astrocytes via application of conditioned culture media. The have used a pool of cytokines to stimulate DRG cultures and then investigated the effect of the conditioned media on spinal glia. They have conducted a well-thought-out series of experiments exploring Ca2+ current sensitization and morphological changes in glia. Quite interestingly they found that media conditioned by DRG cultures following one media change and removal of the cytokines still stimulates a spinal glia response. Future studies would do well to address which soluble factors in the media are being released in the DRG cultures, and by which cell type, that end up stimulating spinal glia. Below, I have split my review into major and minor concerns.
Major Issues
- The authors rationalize the development of this series of experiments as a way to study nociceptor-glia interaction in vitro and why this could be useful for chronic pain conditions, such as chronic low back pain associated with disc degeneration. With this in mind, it is unclear why they used neonatal rat derived DRG cultures and spinal glia cultures. The nervous system is still developing in neonatal rats and the nociceptive system also changes with age (for example see PMID: 26866058). Furthermore, NGF was added to cultures, presumably because neonatal neuronal cultures are not viable without NGF. NGF however is elevated in chronic pain conditions like disc degeneration and osteoarthritis, so its addition could make these conditions difficult to study in the in vitro system that the authors have presented. NGF could also have a priming effect on nociceptors, thus skewing the results when an actual stimulus is added and making it challenging to interpret the results (ie. Is the effect of factor x due to factor x, or does factor x only have the observed effect because of NGF priming?). Using cultures derived from adult rats or mice could increase the clinical relevance of this in vitro system and also remove the need for NGF trophic support.
- Much of the data should be analyzed in a statistically more rigorous manner. For most experiments the authors have used a biological/experimental n=3, with a number of technical replicates (often fields of view) for each experiment. The authors then used the technical replicates, rather than the experimental replicates, for their statistical analyses such as in Fig. 2D, 3G,H, K and L, 4C to F and so on. This approach treats technical replicate as an individual experiments, which they are not (two pictures of cells in the same well can definitely not be considered different experiments), and this will give the statical testing give more degrees of freedom and power, artificially increase the n, and making it easier to reach statistical significance. In addition to the approach the authors have taken, they should calculate the mean or median for each experiment and use those three data points for their analyses. It is also unclear in how the statistical testing was conducted in the ‘variance among donors’ graphs just as in 2E. Was the testing done with one data point per donor generated from the technical replicates or was it done with all of the technical replicates from each donor?
- There are many cell types in the dorsal root ganglia besides neurons and many cell types besides microglia and astrocytes in the spinal cord that could influence the results. For example, stimulating DRG cultures with the mix of cytokines used could directly stimulate neurons and cause them to signal to other cells in the culture, or vice versa. This in turn will have an effect when the condition media is added to the spinal glia cultures. For example, satellite glia cell derived cytokines or other soluble factors that would not normally reach the spinal cord would be released into the conditioned media and then added to the spinal glia cultures. The better understand this caveat the authors should at least determine the purity of the cultures (i.e. the percent of all cells in each culture that are neurons, astrocytes or microglia) and it would be best if they determine the other cell types in their cultures.
Minor issues
Introduction
- The authors discuss the role of inflammatory cytokines in disc degeneration and animal models of disc degeneration and they separately discuss glia activation in pain models. They could additionally discuss the link between disc degeneration, cytokines and glia activation in pre-clinical models of disc degeneration and chronic low back pain (see PMID: 30326776, 31047862, 24718079, 26394856)
Results
- Please define what is in the cytokine cocktail in the text of the results section (line 81)
- CM is used as an acronym for both conditioned media in general, and the control conditioned media that did not have cytokines. I find this slightly confusing. Perhaps using two distinct acronyms or just spelling out ‘conditioned media’ would be clearer.
- Was the conditioned media frozen between collection and application to the mixed glia cultures? The freeze thaw cycle could impact different soluble factors to varying degrees. While it is avoidable, it is an important detail to mention.
- Looking at Figure 3, the GFAP signal looks much more intense than CD11b. What strategy did the authors use to determine their laser and gain settings while imaging their cultures? This can affect any downstream intensity analyses.
- Line 286 – should ‘astrocytic calcium’ response be ‘microglia’?
Discussion
- The authors should discuss the caveats associated with a conditioned media culture system not recapitulating the fact that only a very small part of a sensory neuron is anywhere near spinal glia whereas in a culture, the entire neuron (along with all the other cells) can release factors into the conditioned media.
Methods
- The authors should consult the ARRIVE guidelines and add additional information concerning to animal housing and conditions.
- What sex were the rats? Recent manuscripts highlighted sex difference in DRG neurons (PMID 32943709, https://www.sciencedirect.com/science/article/pii/S0006322320319521)